# On Provable Benefits of Policy Learning from Human Preferences in Contextual Bandit Problems

## Abstract

For a real-world decision-making problem, the reward function often needs to be engineered or learned. A popular approach is to utilize human feedback to learn a reward function for training. The most straightforward way to do so is to ask humans to provide ratings for state-action pairs on an absolute scale and take these ratings as reward samples directly. Another popular way is to ask humans to rank a small set of state-action pairs by preference and learn a reward function from these preference data. Recently, preference-based methods have demonstrated substantial success in empirical applications such as InstructGPT. In this work, we develop a theoretical comparison between these human feedback approaches in offline contextual bandits and show how human bias and uncertainty in feedback modelings can affect the theoretical guarantees of these approaches. Through this, our results seek to provide a theoretical explanation for the empirical successes of preference-based methods from a modeling perspective.

## 1 Introduction

Reward engineering is one of the most crucial aspects in real-world decision-making problems. It is particularly important to bandits and reinforcement learning (RL), since it can be prohibitively expensive to learn the entire environment through random exploration and most existing algorithms rely on a reward function to guide their exploration in a deliberate manner so that they can solve the desired tasks efficiently.

In some cases, it can be straightforward to select reward functions when we have sufficient prior knowledge about the dynamics or rules that govern the problems of interest, such as games and simulated physical systems [34; 35; 50]. However, this is often not the case in practice. Real-world problems can be highly complex, and there may not be a clear choice of reward to use. Therefore, practitioners often have to construct a reward function from scratch for their algorithms to use. Unfortunately, such artificial rewards often end up misaligned with the overall objective of the system and fail to lead to the desired policy. For instance, in the task of teaching a chatbot to converse like a human, assigning a scalar reward to a chatbot's reply is challenging since there is no scale that can objectively evaluate its quality. Therefore, reward engineering poses a significant challenge to policy learning, particularly when it is difficult to quantify policy performance or the system is multi-objective.

To address these challenges, a popular methodology is to learn the reward function from human feedback instead of handcrafting it from scratch. These methods assume a true reward function exists and its corresponding optimal policy is aligned with our goal, but this true reward is not accessible or directly observable. Instead, it needs to be learned with the feedback data from human annotators, who are believed to be able to evaluate an algorithm or agent in a way aligned with the true reward. The most straightforward approach is to ask human annotators to rate subjects on an absolute scale [29; 30; 20]. These ratings can be either used directly as reward samples or incorporated into a pre-designed reward function as a component [53]. We refer to this use of human feedback as "rating". The rating approach has been popular because of its easy implementation and the compatibility of such rating data with most existing algorithms. However, as some empirical studies have shown [58; 24], the reward derived from human ratings is susceptible to bias and uncertainty of the human annotators and can deviate from the true reward. To characterize the ratings given by human

annotators, several models have been proposed and widely adopted in the literature [40; 31]. These existing models, however, fall short on two aspects: (i) they model the uncertainty noise in a simple, isolated form, and (ii) their modeling of the bias is also restrictive, which does not fully capture the bias in practice and can render the problem of policy learning statistically inconsistent.

As an alternative, there has been a growing trend in recent years to use human feedback by comparing subjects rather than rating them individually. These methods are commonly referred to as preference-based methods. In lieu of assigning ratings on an absolute scale, human annotators are given small sets of subjects and tasked with comparing and ranking the subjects within each set. Since some empirical studies have shown that humans are fairly accurate when making choices among a small number of subjects [36; 52; 67], the preference-based approach is believed to be more robust to human bias and uncertainty and learn reward more accurately. The preference feedback of human annotators is commonly modeled with either the Bradley-Terry-Luce model [6] or the Plackett-Luce model [41], both of which have found extensive applications in recommender systems and crowdsourcing research [10; 47; 15]. Recently, preference-based methods have become popular for reward learning in bandit problems and have played a crucial role in the remarkable success of training large language models such as InstructGPT and ChatGPT.

While the preference-based approach has demonstrated notable empirical effectiveness in reward engineering, its theoretical properties remain largely unexplored. Existing results have primarily focused on algorithms for online bandit and RL, whose goal is to maximize a preference metric rather than to learn a reward function [39; 65]. Recently, [80; 78] proved the optimal policy can be learned from preference data in the offline setting with pessimism and maximum likelihood estimation (MLE) and analyzed the suboptimality. [57] showed any robust RL algorithm can be adapted to find the optimal policy with preference data, suggesting preference-based policy learning is no harder than standard robust RL. However, the reason why the preference-based approach outperforms the traditional rating-based approach in practice still remains a question. In this work, we provide a theoretical comparison between these human feedback approaches and propose a theory that aims to explain the advantage of the preference-based approach over rating in policy learning from a modeling perspective. To align with recent applications [13; 38], we focus on tabular contextual bandits in the offline setting.

Specifically, we first consider a new model for human rating data and analyze the suboptimality guarantees of the standard LCB algorithm under it. Our rating model is based on a general class of monotone functions that can account for both human bias and uncertainty with general forms. By incorporating the concept of monotonicity, our model captures the bias observed in real-world human rating and maintains the correct reward ordering. This allows policy learning to be a consistent yet nontrivial statistical problem, which differs from existing rating models that do not preserve the reward ordering or guarantee the consistency of the induced policy learning problem. In addition, our model is able to express a more general form of noise to represent human uncertainty during rating. Through our models, we provide the first known suboptimality analysis for reward engineering with human rating in bandit problems and shed light on how human bias and uncertainty can adversely impact policy learning. Furthermore, we compare our results with the suboptimality result from [80] for the preference-based method pessimistic MLE. The comparison reveals that the preference-based approach enjoys lower suboptimality than the rating-based approach when human bias is extreme in human rating. Lastly, we also consider a new model for human preference with human bias and compare the sample complexity of pessimistic MLE under this new model with the results for human rating. This comparison shows when human bias and uncertainty are equally strong in both types of human feedback, the preference-based approach has no provable advantage over the rating-based one. Altogether, our theory shows the advantage of the preference-based approach can be largely attributed to its modeling with mild human bias and uncertainty, which makes it reasonable to believe the great empirical success of preference-based methods is because human preference data is subject to less bias and uncertainty in practice.

## 1.1 RELATED WORKS

**Preference-based reinforcement learning.** Preference-based Reinforcement Learning (PbRL) [13; 49; 8; 59; 60; 14; 1] has been studied under different types of human feedback including action comparison, state comparison and trajectory comparison—see [60; 1] for reviews of the literature. Preference-based feedback has been well-studied in a bandit setting known as dueling bandits [2; 5; 17; 33; 44; 45; 46; 73; 74; 75; 81] with a survey in [4]. Recently, there is a growing interest in

the theoretical guarantees of PbRL methods, including tabular case [37; 65] and linear and general function approximations [39; 11; 79]. However, these works are focused on the online setting and their methods are not applicable to the offline setting.

**Offline policy learning.** The vast literature on offline policy learning can be divided by the different assumptions on the data sampling distribution. The strongest assumption is one that requires all state-action pairs can be sampled [70; 16; 9; 61]. A similar assumption requires that the occupancy measure of every possible policy be dominated by the data sampling distribution, which is common in the function approximation setting [3; 51]. One of the weakest assumptions is one that requires the occupancy measure of an optimal policy be dominated by the data sampling distribution. To learn the optimal policy under this setting, the principle of pessimism [25; 7; 23] was introduced and has inspired many algorithms [72; 42; 28; 63; 76; 77; 62; 64]. In particular, the sample complexity of pessimistic algorithms has been extensively studied in the tabular case [48; 66; 27; 68; 43; 63; 69; 42] and linear MDPs [23; 62; 77; 56; 18; 71]. In this spirit, the algorithms we study in this work also use human feedback with pessimism.

**Notation**   Given any vector $x \in \mathbb{R}^{SA}$ that represents a function $x : \mathcal{S} \times \mathcal{A} \to \mathbb{R}$, we use $x(s, a)$ to denote the entry corresponding to the state-action pair $(s, a)$. For any random sample $X$ with distribution $P$ and a function $f(\cdot)$ of $X$, we denote the expectation of $f(X)$ over $P$ with $\mathbb{E}_{X \sim P}[f(X)]$ or $\mathbb{E}_X[f(X)]$. Similarly, we denote the variance of $f(X)$ over $P$ with $\mathrm{Var}_X(f(X))$. Lastly, we denote equality in distribution with $\stackrel{d}{=}$.

## 2   PRELIMINARIES

In this section, we make a brief review of the contextual bandit problem and policy learning in the offline setting.

### 2.1   CONTEXTUAL BANDIT

We consider a contextual bandit represented by $(\mathcal{S}, \mathcal{A}, r, \rho)$. Specifically, we focus on a tabular setting, where $\mathcal{S} := 1, 2, \ldots, S$ denotes the state space of size $S$, and $\mathcal{A} := 1, 2, \ldots, A$ denotes the action space of size $A$. The function $r : \mathcal{S} \times \mathcal{A} \to [0, R]$ represents the true reward function, which is assumed to be deterministic and unknown in this paper. Here, $r(s, a)$ is the immediate reward obtained when taking action $a \in \mathcal{A}$ in state $s \in \mathcal{S}$. $\rho$ denotes the initial state distribution of the bandit.

A policy $\pi : \mathcal{S} \to \Delta(\mathcal{A})$ specifies how actions are selected given a state, where $\pi(\cdot|s) \in \Delta(\mathcal{A})$ represents the action selection probability vector at state $s \in \mathcal{S}$. We also use $\pi(s)$ to denote the action selected by policy $\pi$ at state $s$. We denote the state-action visitation distribution of $\pi$ starting from the initial distribution $\rho$ with $d_\rho^\pi$. The value function of policy $\pi$ is defined as follows:

$$V^\pi(s) := \mathbb{E}_{s \sim \rho, a \sim \pi(\cdot|s)}\left[r(s, a)\right].$$

Lastly, an optimal policy $\pi^\star$ maximizes the value function for all states simultaneously.

### 2.2   OFFLINE POLICY LEARNING

We consider the offline setting of policy learning, in which the learner is given a dataset of i.i.d. samples generated under a sampling distribution. While the sampling distribution can take different forms under different feedback models, the task is always to learn a policy $\pi$ from the offline dataset that performs as well as the optimal policy as possible. In particular, we evaluate the performance of $\pi$ by the suboptimality metric defined as follow:

$$\mathrm{SubOpt}(\pi) := \mathbb{E}_{s \sim \rho}\left[V^{\pi^\star}(s) - V^\pi(s)\right]. \tag{1}$$

Here the suboptimality measures the performance difference between the optimal policy $\pi^\star$ and $\pi$ in the problem bandit. Naturally, one aims to minimize the suboptimality and find an algorithm whose suboptimality converges to zero as the sample size $n$ approaches infinity.

## 3   HUMAN RATING MODELS

One of the most straightforward ways to use human feedback is to let human annotators evaluate on an absolute scale. Since such data can be readily used in most algorithms in the existing literature, the human rating approach has become very popular and one of the most important to study. As

evidenced in behavioral studies, human ratings are subject to both bias and uncertainty [20; 32]. Specifically, under the influence of their own personalities and past experiences, human annotators can exhibit personal opinion bias during rating, leading to deviations from the true score. Furthermore, due to the tedious and challenging nature of the rating process, the evaluation of the same subject by the same human annotator can fluctuate over time, giving rise to what is known as intra-observer uncertainty. In light of these phenomena, [20] propose a model that aims to characterize the ratings from a human annotator in the real world, which has been widely adopted in the existing literature [40; 31; 29]. In the following, we present this model in the single annotator case under the contextual bandit setting. For any fixed state-action pair $(s, a) \in \mathcal{S} \times \mathcal{A}$ with true reward $r(s, a)$, a rating from human annotator $\widetilde{r}(s, a)$ can be written as

$$\widetilde{r}(s, a) = r(s, a) + \Delta_{(s,a)} + \epsilon. \tag{2}$$

Here, $\Delta_{(s,a)}$ represents the bias of the human annotator for action $a$ at state $s$. Learners and algorithms have no knowledge of such bias and would take these observed ratings as reward samples directly. In most works [20, 32], $\Delta_{(s,a)}$ is modeled as an unknown constant; in [40], $\Delta_{(s,a)}$ is a Gaussian random variable with an unknown, non-zero mean. $\epsilon$ is a random noise representing the intra-observer uncertainty, which is modeled with a zero-mean Gaussian random variable in these works.

Apparently, such a human rating model bears several limitations. In (2), the bias $\Delta_{(s,a)}$ has an unknown non-zero expectation, which makes it impossible to recover the true reward $r(s, a)$ exactly. Furthermore, identifying the optimal action for a given state $s$ becomes infeasible when the bias causes a flip in the expected reward, i.e, $\mathbb{E}[\widetilde{r}(s, a)] > \mathbb{E}[\widetilde{r}(s, \pi^\star(s))]$ for any $a \notin \pi^\star(s)$, in which case the policy learning problem becomes inconsistent under this model. However, in the real world, the bias of a human annotator should not keep him or her from identifying the best action in expectation, but it is likely to take a much more general form. Neither of these is reflected in the current modeling with additive constant bias. On the other hand, the uncertainty is only modeled with an additive sub-gaussian random variable in these works. While this simple noise model is considered in many theoretical works, it cannot capture the setting when reward samples are generated from human rating. In practice, such uncertainty noise can be higher-order and more complex [22]. In addition, the uncertainty might also be affected by the bias and reward at different state-action pairs and have an intricate role in the final observation $\widetilde{r}(s, a)$.

To model human rating more realistically while keeping the policy learning problem consistent, we propose a new model under which the rating $\widetilde{r}(s, a)$ can be expressed with

$$\widetilde{r}(s, a) = h(r(s, a), \epsilon), \tag{3}$$

where $h(\cdot, \cdot)$ is a general, deterministic transformation and $\epsilon$ is a random variable sampled from $\mathcal{N}(0, \sigma^2)$ and independent from $(s, a)$. For notational simplicity, we define $\bar{h}(r) := \mathbb{E}_\epsilon[h(r, \epsilon)]$. This can be interpreted as the reward function for this bandit instance in the human annotator's mind, which he or she uses to produce ratings. We can refer to $\bar{h}(r)$ as the biased reward and the function $\bar{h}(\cdot)$ as the expected bias function.

While the $h$ transformation can be general in our model, for the policy learning problem to make sense, it needs to satisfy three conditions. This can be viewed as the set of our assumptions on human rating feedback, which we theorize captures the real-world scenario accurately. In particular, we only consider models that satisfy these conditions:

**Condition 1.** The function $\bar{h}(\cdot)$ satisfies

$$\bar{h}(r_1), \bar{h}(r_2) \in [0, R] \quad \text{and} \quad \bar{h}(r_1) > \bar{h}(r_2)$$

for any $r_1, r_2 \in [0, R]$ such that $r_1 > r_2$. In addition, $\bar{h}(0) = 0$.

This condition assumes that $\bar{h}(\cdot)$ is strictly monotone, implying that the biased reward function should preserve the ranking under the true reward $r$ in expectation. This condition is particularly important, as it ensures that the consistency of the policy learning problem. Therefore, we can identify the optimal policy based on human rating data in expectation.

**Remark 1.** The monotonicity in Condition 1 also guarantees that any group of samples can be correctly compared and ranked in expectation, which is a necessary condition for the use of preference-based methods. This unifies the admissibility assumption in the rating models and preference models, which is crucial for the validity of our subsequent theoretical comparison between the two approaches.

**Remark 2.** We require $\bar{h}(0) = 0$ only to rule out arbitrary constant shift in $\bar{h}$ because shifting the reward by a constant is trivial and does not change the policy learning problem or any theoretical guarantee.

**Condition 2.** For any $r \in [0, R]$, $h(r, \epsilon)$ is a degree-$q$ polynomial in $\epsilon$ and symmetric in $\epsilon$ about its expectation, i.e.,

$$-(h(r, \epsilon) - \mathbb{E}_\epsilon[h(r, \epsilon)]) \overset{d}{=} h(r, \epsilon') - \mathbb{E}_{\epsilon'}[h(r, \epsilon')],$$

where $\epsilon'$ is a random variable identically distributed as $\epsilon$.

Since $h(r, \cdot)$ can be a very general function, the human uncertainty noise in the final observation $\widetilde{r}(s, a)$ is allowed to have a complicated dependency on the bias and the true reward, even though the randomness only comes from an independent Gaussian random variable $\epsilon$. For instance, the original white noise $\epsilon$ may end up amplified or reshaped by the true reward and human's internal bias and cause the final ratings to exhibit a complex concentration around $\bar{h}(r)$. This not only provides more realism and flexibility in modeling but also presents a greater theoretically challenge compared to the uncertainty considered in (2), which is modeled with a simple additive Gaussian noise with no interactions with the true reward and human bias.

**Remark 3.** Condition 2 is only a regulation on the effect of uncertainty—the uncertainty noise should not favor any particular direction (though the bias still can). This is in line with the real world, where the random noise concentrates evenly around the expectation and the effect of uncertainty diminishes in expectation.

**Condition 3.** For any $r_1, r_2 \in [0, R]$ such that $r_1 \geq r_2$, there are positive constants $C_{h,1}, C_{h,2} > 0$ such that

$$\bar{h}^{-1}(r_1) - \bar{h}^{-1}(r_2) \leq C_{h,1} \cdot \bar{h}^{-1}(r_1 - r_2) \quad \text{and} \quad \bar{h}(r_1) - \bar{h}(r_2) \leq C_{h,2} \cdot \bar{h}(r_1 - r_2).$$

This is a technical condition on the regularity of the expected bias function. It ensures that the bias does not transform the reward too drastically, which eases our theoretical analysis.

Overall, this $h$ transformation can model very general behavior. For example, many human annotators with strong personal opinions tend to exhibit an extreme bias in their evaluations, making them rate subjects with low true reward even lower and rate those with high true reward even higher on average. Our model can capture such bias with a convex $\bar{h}(\cdot)$, with a concrete example and its theoretical guarantees detailed in Appendix B.

## 4    RESULTS FOR HUMAN RATING

Before the theoretical comparison with the preference-based approach, let us first establish some theoretical results for our more general rating model. In particular, we analyze the suboptimality of the LCB algorithm under our more practical rating model. These results can provide some theoretical explanation for how human bias and uncertainty could adversely affect policy learning.

In the case of human rating, we are given an offline dataset $\mathcal{D} = \{(s_i, a_i, \widetilde{r}_i)\}_{i=1}^n$. The state-action pairs in $\mathcal{D}$ are generated in an i.i.d. fashion according to a sampling distribution over the state-action space. The sampling probability of the state-action pair $(s, a)$ is denoted with $d(s, a)$. For each $(s_i, a_i)$, the human annotator provides a *rating sample* $\widetilde{r}_i$ following the rating model (3) based on the true reward $r(s_i, a_i)$.

Let us also make a brief review of the standard LCB approach for offline policy learning [23; 42; 69]. In the existing literature, it is common to assume the knowledge of a reasonable upper bound on the variance of reward observations. Similarly, we assume there exists an upper bound on the variance $\text{Var}_\epsilon(h(r, \epsilon))$ for all $r \in [0, R]$, which we denote with $V_{R,\sigma}^2$ and can depend on $R$ and $\sigma$. Recall that the learner has no knowledge of the transformation $h$, but let us assume the learner can make a reasonable estimate $\widetilde{V}_{R,\sigma}^2$ for the true variance $V_{R,\sigma}^2$ such that $\widetilde{V}_{R,\sigma}^2 = c_V V_{R,\sigma}^2$, where $c_V > 0$ is an absolute constant. To learn the optimal policy with at least $1 - \delta$ success probability, the standard LCB algorithm (Algorithm 1) uses a penalty in the form of

$$b_m = c_b \sqrt{\frac{\widetilde{V}_{R,\sigma}^2 \log \frac{SA}{\delta}}{m}} \tag{4}$$

with an appropriately chosen constant $c_b$.

---

**Algorithm 1:** LCB for contextual bandits

---

1 **Input:** Offline rating dataset $\mathcal{D}$, confidence level $\delta \in (0, 1)$.
2 **for** *all* $(s, a) \in \mathcal{S} \times \mathcal{A}$ **do**
3    Set $n_{(s,a)} = \sum_{i=1}^{n} \mathbb{1}\{(s_i, a_i) = (s, a)\}$;
4    Set $\widetilde{r}(s, a) = \frac{1}{n} \sum_{i=1}^{n} \widetilde{r}_i \mathbb{1}\{(s_i, a_i) = (s, a)\}$;
5    Set $\widehat{r}(s, a) = \max\{\widetilde{r}(s, a) - b_{n_{(s,a)}}, 0\}$;
6 **return** $\widehat{\pi}_{\mathrm{LCB}}(\cdot) = \arg\max_{a \in \mathcal{A}} \widehat{r}(\cdot, a)$.

---

To understand the effects of human bias and uncertainty on policy learning under our more realistic rating model, let us establish the lower bound on the suboptimality of the LCB algorithm. We will consider two scenarios with different coverage assumptions for the offline dataset $\mathcal{D}$.

### 4.1 LOWER BOUND UNDER PARTIAL COVERAGE

As [42; 69] have shown, to learn the optimal policy in the offline setting, it is sufficient for the sampling distribution of the offline dataset to cover the state-action pairs that the optimal policy can reach. Concretely, this assumption can be written as follows:

**Assumption 1.** *There exists an optimal policy $\pi^\star$ such that $d(s, a) > 0$ whenever $d_\rho^{\pi^\star}(s, a) > 0$ for any $(s, a) \in \mathcal{S} \times \mathcal{A}$.*

Under this assumption, it makes sense to define a concentrability coefficient $C^\star$ as follows:

$$C^\star := \max_{(s,a) \in \mathcal{X}} \frac{d_\rho^{\pi^\star}(s, a)}{d(s, a)}, \tag{5}$$

where the set $\mathcal{X}$ is the set of all state-action pairs that the sampling distribution of $\mathcal{D}$ can cover, i.e., $\mathcal{X} := \{(s, a) \in \mathcal{S} \times \mathcal{A} : d(s, a) > 0\}$. Under Assumption 1, if the reward can be observed exactly or with only additive sub-gaussian noise, the LCB algorithm (Algorithm 1) with penalty (4) is guaranteed to converge to the optimal policy [42; 69]. However, theory suggests it does not converge in the worst case when the reward function is engineered from human rating. In particular, let us consider the setting beyond the standard additive sub-gaussian noise, which has been well-studied in the existing literature. That is, let us consider a more practical model in the form of (3) with $q \geq 2$. We can prove that even when the rating model preserves the correct reward ordering in expectation and keeps the policy learning problem consistent, it is possible that the LCB algorithm does not converge to the optimal policy and must suffer constant suboptimality.

**Theorem 1.** *For any fixed constant $0 < \delta < 1$, there exists a contextual bandit instance with initial state distribution $\rho$ such that if one samples a dataset $\mathcal{D}$ of size $n \geq c(\delta, c_b, c_V, q, \sigma, R)$ using a sampling distribution $d$ satisfying Assumption 1 with $C^\star = 2$ and runs Algorithm 1 on $\mathcal{D}$, the output policy $\widehat{\pi}_{\mathrm{LCB}}$ must suffer constant suboptimality, i.e.,*

$$\mathbb{E}_{\mathcal{D}}[\mathrm{SubOpt}(\widehat{\pi}_{\mathrm{LCB}})] = c_0 R, \tag{6}$$

*where $c_0$ is a universal constant and $c(\delta, c_b, c_V, q, \sigma, R)$ is a constant depending on $\delta, c_b, c_V, q, \sigma, R$.*

This result is reminiscent of Proposition 1 in [42], which constructs a bandit and shows the empirically best policy chooses a suboptimal action with constant probability under Assumption 1. The very same work also shows that by adding a pessimism penalty, the LCB algorithm (Algorithm 1) can converge to the optimal policy under the same data coverage assumption. In contrast, our theorem shows that even when we make pessimistic estimates and penalize less-observed state-action pairs in human rating data, a constant suboptimality can still ensue. This shows a disadvantage of using human rating as reward samples directly: although the estimation problem induced by human rating is still consistent, using LCB with only the knowledge of variance is not sufficient for convergence. Instead, the learner needs to know the shape of the noise distribution, but it is unrealistic to model the human uncertainty accurately in practice. We also provide an upper bound result for the case when the learner has complete knowledge of the uncertainty noise distribution in Appendix C.

**Proof sketch** In a bandit instance with special reward design, we first find the lower bound for the probability that suboptimal actions are only observed for a very small number of times in the offline dataset. Such state-action pairs can have huge fluctuation in their empirical reward average and mislead the algorithm. Then, we find the lower bound on the probability that a state-action

pair $(s, a)$ such that $\widehat{r}(s, a) > \widehat{r}(s, a^\star)$ exists, which can cause the algorithm to always select the suboptimal action $a$ and suffer suboptimality. Different from Proposition 1 in [42], in which the reward noise for suboptimal actions is defined with two Dirac delta functions, the noise under our rating model is unbounded and can be viewed as a Gaussian chaos, so we compute this probability using a method from the corresponding literature. Moreover, in the same paper, a bandit instance is sufficient to induce constant suboptimality as long as its action space is designed large. In our case, since the pessimism penalty in Algorithm 1 accounts for the bandit size and larger bandit instances are penalized more, it requires a careful balance in the design of our bandit instance.

### 4.2 LOWER BOUND UNDER FULL COVERAGE

Uniform coverage is another popular coverage assumption for offline policy learning [70; 54; 19]. It can be written as follows:

**Assumption 2.** *The sampling distribution satisfies $d(s, a) > 0$ for any $(s, a) \in \mathcal{S} \times \mathcal{A}$.*

This coverage assumption is much stronger than Assumption 1 and makes the offline policy learning problem much easier. Under Assumption 2, many algorithms without the pessimism principle can also be shown to provably converge to the optimal policy [9; 61]. Moreover, [23] showed that the suboptimality of algorithms with pessimism can decay faster when the data are well-explored. In this setting, we establish a lower bound on the suboptimality of Algorithm 1 under Assumption 2.

**Theorem 2.** *For any fixed constant $0 < \delta < 1$, there exists a contextual bandit instance with initial state distribution $\rho$ such that if one samples a dataset $\mathcal{D}$ of size $n \geq \max\{48\sigma^4, 60\}$ using a sampling distribution $d$ satisfying Assumption 2 with $d(s, a) = \frac{1}{SA}$ for every $s \in \mathcal{S}$ and $a \in \mathcal{A}$ and runs Algorithm 1 on $\mathcal{D}$, the output policy $\widehat{\pi}_{\mathrm{LCB}}$ must suffer suboptimality at least*

$$\mathbb{E}_{\mathcal{D}}[\mathrm{SubOpt}(\widehat{\pi}_{\mathrm{LCB}})] = c_0 \cdot \bar{h}^{-1}\left(\sqrt{\frac{V_{R,\sigma}^2}{n}}\right),$$

*where $c_0$ is a constant that depends on $q$.*

In fact, under uniform data coverage as in Theorem 2, pessimism becomes unnecessary and this result holds no matter what penalty $b_n$ is used in the algorithm. This theorem demonstrates another disadvantage of human rating: even when the data covers the entire state-action space and learning is no longer impeded by the lack of knowledge of human uncertainty, the suboptimality is still bottlenecked by human bias.

## 5 COMPARISON WITH PREFERENCE-BASED APPROACH

In contrast to rating, the preference-based approach relies on models that characterize how a human annotator would rank a group of subjects by reward. In this case, the feedback is simply the most preferred subject to the human annotator within the group. Such feedback actually contains less information than rating. Preference data are also incompatible with standard bandit algorithms and require special adaptation to use [57]. However, the preference-based approach has received much attention recently because some have found it easier and more accurate for human to make preferences than rating [36; 52; 67]. In this section, we compare the human rating approach with the preference-based approach.

### 5.1 HUMAN PREFERENCE UNDER BTL

Let us consider the most basic case of human preference called pairwise comparison, which involves the ranking between a pair of state-action pairs based on their rewards. This is predominantly modeled with the Bradley-Terry-Luce (BTL) model [6], under which a human annotator gives a binary response $y = \{0, 1\}$ following a Bernoulli distribution when asked to compare two state-action pairs $(s, a^0)$ and $(s, a^1)$ with $a^0 \neq a^1$:

$$P(y|s, a, a') = \frac{\exp(r(s, a^y))}{\exp(r(s, a^0)) + \exp(r(s, a^1))}. \tag{7}$$

Like our rating model in (3), the BTL model admits a consistent statistical problem. The learner is given a dataset $\mathcal{D}' = \{(s_i, a_i^0, a_i^1, y_i)\}_{i=1}^n$, which contains i.i.d. human preference samples from some sampling distribution. $y_i$ is the binary human preference feedback for the comparison between $(s_i, a_i^0)$ and $(s_i, a_i^1)$. We denote the sampling probability of the state-action-action triplet $(s, a^0, a^1)$ with $d(s, a^0, a^1)$.

---

**Algorithm 2:** Pessimistic MLE for contextual bandits

---

1 **Input:** Offline preference dataset $\mathcal{D}'$, confidence level $\delta \in (0,1)$.

2 Construct the reward function set $\mathcal{F} := \{v \in \mathbb{R}^{SA} : \mathbf{1}^\top v = 0, \|v\|_\infty \leq R\}$;

3 Set

$$\widetilde{r} = \arg\max_{f \in \mathcal{F}} \sum_{i=1}^n \log\left(\frac{\mathbb{1}\{y_i = 1\}\exp(f(s_i, a_i^1))}{\exp(f(s_i, a_i^0)) + \exp(f(s_i, a_i^1))} + \frac{\mathbb{1}\{y_i = 0\}\exp(f(s_i, a_i^0))}{\exp(f(s_i, a_i^0)) + \exp(f(s_i, a_i^1))}\right);$$

4 Construct empirical covariance matrix

$$\widehat{\Sigma} = \frac{1}{n}\sum_{i=1}^n \left(\mathbf{1}_{(s_i,a_i^0)} - \mathbf{1}_{(s_i,a_i^1)}\right)\left(\mathbf{1}_{(s_i,a_i^0)} - \mathbf{1}_{(s_i,a_i^1)}\right)^\top;$$

5 Construct the pessimistic reward function set

$$\mathcal{F}_{\mathrm{CR}}(\widetilde{r}) = \left\{f \in \mathcal{F} \ : \ \sqrt{(f - \widetilde{r})^\top \widehat{\Sigma}(f - \widetilde{r})} \leq b_n'\right\};$$

6 **return** $\widehat{\pi}_{\mathrm{PMLE}} = \arg\max_\pi \min_{\widehat{r} \in \mathcal{F}_{\mathrm{CR}}(\widetilde{r})} \mathbb{E}_{s \sim \rho}[\widehat{r}(s, \pi(s))]$.

---

To find the optimal policy with human preference data, we can use pessimistic MLE [80], which first computes a reward function by MLE and then outputs the optimal policy corresponding to a pessimistic version of this MLE reward (Algorithm 2). The data coverage assumption is similar to Assumption 1, which essentially requires the sampling distribution to covers the state-actions pairs that optimal policy can reach. In the tabular case, this assumption can be written as follows:

**Assumption 3.** *There exists an optimal policy $\pi^\star$ such that the pairwise concentrability coefficient*

$$C^\dagger := \sqrt{\sup_{v \in [-1,1]^{SA} : \mathbf{1}^\top v = 0} \frac{\left(\sum_{(s,a)} d_\rho^{\pi^\star}(s,a)v(s,a)\right)^2}{\sum_{(s,a^0,a^1)} d(s,a^0,a^1)\left(v(s,a^0) - v(s,a^1)\right)^2}} \tag{8}$$

*is bounded.*

[80] proved the convergence of pessimistic MLE in the linear bandit setting. The following theorem is a special case of Theorem 3.2 from [80] with some modification, which expresses everything in the tabular setting. This shows when we assume human preference follows the BTL model, pessimistic MLE can provably converge to the optimal policy under the mild data coverage assumption of Assumption 3 and its suboptimality decays at a fast rate of $O(1/\sqrt{n})$. This result marks a clear distinction from the negative results for human rating.

**Theorem 3.** *Denote $\gamma = \frac{1}{2 + \exp(R\sqrt{SA}) + \exp(-R\sqrt{SA})}$. Suppose Assumption 3 holds. For any fixed constant $0 < \delta < 1$, if one runs Algorithm 2 with*

$$b_m' = c_b'\sqrt{\frac{SA + \log\frac{1}{\delta}}{\gamma^2 m}},$$

*where $c_b'$ is an appropriately chosen universal constant, with probability $1 - \delta$, the suboptimality of the output policy $\widehat{\pi}_{\mathrm{PMLE}}$ satisfies*

$$\mathrm{SubOpt}(\widehat{\pi}_{\mathrm{PMLE}}) \leq c_0 C^\dagger R\left(\sqrt{\frac{SA + \log\frac{1}{\delta}}{\gamma^2 n}} + \sqrt{\frac{S^2 A^2 \log\frac{n}{\delta}}{n}}\right),$$

*where $c_0$ is a universal constant.*

We can compare the suboptimality in this theorem with the results for the rating-based approach. A comparison with Theorem 1 shows the uncertainty in human ratings may require the data to have stronger coverage in order to converge to the optimal policy. A comparison with Theorem 2 shows when the bias in human ratings distorts the reward function and makes it more extreme and drastic (less smooth in the Lipschitz sense), the $\bar{h}^{-1}(\cdot)$ can slow down the suboptimality's decay with respect to the sample size. In fact, we can observe that the preference-based approach enjoys faster suboptimality decay because preference feedback contains no bias and mild uncertainty

noise according to the BTL model. While such modeling is justified by empirical evidences, it makes one wonder whether the advantage of preference-based methods mostly comes from the modeling aspect. To delve into this further, let us make another theoretical analysis for the case when preference data are affected by human bias.

## 5.2 HUMAN PREFERENCE UNDER BIASED BTL

Let us introduce a new model for human preference called the biased BTL model. This model considers the case when human preferences are also subject to bias just like the rating model (3) and the feedback is generated with respect to the biased reward. In particular, the binary feedback $\widetilde{y} = \{0, 1\}$ for $(s, a^0)$ and $(s, a^1)$ follows:

$$P(\widetilde{y}|s, a, a') = \frac{\exp(\bar{h}(r(s, a^{\widetilde{y}})))}{\exp(\bar{h}(r(s, a^0))) + \exp(\bar{h}(r(s, a^1)))}, \tag{9}$$

where $\bar{h}$ is the expected bias function from (3).

We consider the performance of pessimistic MLE (Algorithm 2) again with human preference data generated under this model. While the data are generated under human bias this time, we still run pessimistic MLE on the new data as before. Different from the suboptimality results in the previous section, we focus on the sample complexity for learning the optimal policy. We take a gap-dependent approach in our analysis to consider the case when human bias closes the biased optimality gap $r(s, \pi^\star(s)) - r(s, a)$ and the actual optimality gap $\bar{h}(r(s, \pi^\star(s))) - \bar{h}(r(s, a))$ remains big, where $a$ is the second best action at $s$. This echoes with the type of undesirable bias we considered in the last comparison, which is true when human annotators have more extreme standards at heart. In a simple bandit instance, we can obtain the following result and notice the samples needed to find the optimal policy with the preference-based approach is no less than the samples needed for the rating-based approach.

**Theorem 4.** *Consider any single-state bandit instance with $\mathcal{A} = \{a_1, a_2\}$ and $0 \leq \bar{h}(r(a_1)) < \bar{h}(r(a_2)) \leq 1$. For any fixed constant $0 < \delta < 1$, let $n_{rate}$ be the total number of samples needed to learn the optimal action with probability at least $1 - \delta$ in the human rating setting under observation model (3) with additive sub-gaussian uncertainty noise and uniform data coverage $n_{a_1} = n_{a_2}$, and let $n_{pref}$ be the number of samples needed to learn the optimal action with probability at least $1 - \delta$ in the human preference setting with observation model (9). It always holds that*

$$\frac{n_{rate}}{n_{pref}} < 0.25\sigma^2. \tag{10}$$

We can see that when the variance proxy of the uncertainty noise $\sigma^2$ is no larger than $4$ in human rating (the expected reward is bounded in $[0, 1]$), the samples needed in the rating-based approach is always fewer than the preference-based approach. This shows if one assumes a similar amount of human bias and uncertainty in both types of human feedback, the preference-based approach is no more sample-efficient. This actually contradicts with the empirical observations in the existing literature, which suggests preference-based methods have superior performance. Hence, our theory shows the bias-free modeling plays a great role in the lower sample complexity of preference-based methods, and our theoretical results can conversely confirm the standard BTL modeling of human preference feedback—it is reasonable to believe human preference data is indeed subject to less bias and uncertainty in practice.

## 6 CONCLUSION

In this work, we have studied policy learning using human feedback for reward engineering in bandit. Specifically, we have provided a theoretical comparison between human rating methods and preference-based methods, which shows human bias and uncertainty can have considerable adverse effect on policy learning. Our theory also suggests the preference-based approach has no provable advantage over the traditional rating-based approach when the two types of human feedback are modeled with equally strong human bias and uncertainty. This implies the reason for the empirical success of preference-based methods might be that human preference data are subject to milder human bias and uncertainty. Beyond this work, it is still open for future work to investigate the case when the human feedback is generated from a mixture model representing a group of annotators and provide a comparison between rating methods and preference-based methods in this setting.

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
