## A  ADDITIONAL NOTATION

Let $f : \mathbb{R} \to \mathbb{R}$ and $g : \mathbb{R} \to \mathbb{R}$ be two functions. We denote the their composition $f(g(\cdot))$ with $(f \circ g)(\cdot)$. We use $\mathcal{N}(\mu, \sigma^2)$ to denote a Gaussian distribution with mean $\mu$ and variance $\sigma^2$. For a probability event $\mathcal{E}$, we denote its complement event with $\mathcal{E}^C$. For a vector $v$, $\|v\|_2$ denotes the $\ell_2$-norm of the vector $v$. For a positive semidefinite matrix $A$, $\|v\|_A$ denotes a semi-norm of the vector $v$ with respect to the matrix $A$ with $\|v\|_A = \sqrt{v^\top A v}$. We denote the set of all positive integers with $\mathbb{Z}_{>0}$.

## B  AN EXAMPLE OF HUMAN RATING MODEL

Consider a setting in which the true reward function satisfies $r(s, a) \in [0, 1]$ for all $s \in \mathcal{S}$ and $a \in \mathcal{A}$. For any $r$,
$$h(r, \epsilon) = r^2 + r^2 \epsilon |\epsilon| \quad \text{and} \quad \bar{h}(r) = r^2. \tag{11}$$
This specific rating model has an interpretation that aligns with how human annotators give ratings in practice. Many human annotators with strong personal opinions and taste tend to exhibit an extreme bias in their evaluations, making them rate subjects with low true reward even lower and rate those with high true reward even higher on average. This is captured by the quadratic form of $\bar{h}(\cdot)$. On the other hand, when the white noise $\epsilon$ has a large magnitude, the impact of uncertainty on the final rating should be even larger and more drastic. Since annotators tend to be more deliberate yet more uncertain about subjects with large true reward, the fluctuation of the final rating should also increase with the magnitude of the true reward of a subject. Note that the uncertainty $r^2 \epsilon |\epsilon|$ is a subexponential random variable and actually not a polynomial of $\epsilon$; however, our theory still applies, since the tail of $r^2 \epsilon |\epsilon|$ is dominated by the quadratic $r^2 \epsilon^2$. More generally, our theory is applicable to any uncertainty noise whose tail is dominated by a degree-$q$ polynomial of $\epsilon$. Furthermore, one can easily check that this $h$-function satisfies all three conditions in Section 3.

Now, let us also provide some corollaries for the suboptimality of the LCB algorithm (Algorithm 1) under the example rating model in (11). In the partial coverage setting under Assumption 1, we have the following corollary.

**Corollary 1.** *For any fixed constant $0 < \delta < 1$, there exists a contextual bandit instance with initial state distribution $\rho$ such that if one samples a dataset $\mathcal{D}$ of size $n \geq c(\delta, c_b, c_V, \sigma, R)$ using a sampling distribution $d$ satisfying Assumption 1 with $C^\star = 2$ and runs Algorithm 1 on $\mathcal{D}$, the output policy $\widehat{\pi}_{\mathrm{LCB}}$ must suffer constant suboptimality, i.e.,*
$$\mathbb{E}_{\mathcal{D}}[\mathrm{SubOpt}(\widehat{\pi}_{\mathrm{LCB}})] = c_0, \tag{12}$$
*where $c_0$ is a universal constant and $c(\delta, c_b, c_V, \sigma, R)$ is a constant depending on $\delta, c_b, c_V, \sigma, R$.*

The proof of this corollary is deferred to Appendix E.2.

Let us also provide a corollary for the suboptimality of the LCB algorithm under the same example rating model in the full coverage setting of Assumption 2.

**Corollary 2.** *For any fixed constant $0 < \delta < 1$, there exists a contextual bandit instance with initial state distribution $\rho$ such that if one samples a dataset $\mathcal{D}$ of size $n \geq \max\{48\sigma^4, 60\}$ using a sampling distribution $d$ satisfying Assumption 2 with $d(s, a) = \frac{1}{SA}$ for every $s \in \mathcal{S}$ and $a \in \mathcal{A}$ and runs Algorithm 1 on $\mathcal{D}$, the output policy $\widehat{\pi}_{\mathrm{LCB}}$ must suffer suboptimality at least*
$$\mathbb{E}_{\mathcal{D}}[\mathrm{SubOpt}(\widehat{\pi}_{\mathrm{LCB}})] = \frac{c_0 \sigma}{n^{1/4}},$$
*where $c_0$ is a universal constant.*

The proof of this corollary is deferred to Appendix F.2. This corollary shows the suboptimality can decay more slowly under the influence of human annotator bias.

## C  UPPER BOUND UNDER ASSUMPTION 1 WITH KNOWLEDGE OF NOISE

To compare with the lower bounds for human rating, we prove an upper bound on the suboptimality of the LCB algorithm (Algorithm 1) in the most benign case that the learner has full knowledge of the uncertainty noise distribution of the rating model and design the LCB penalty $b_n$ accordingly. This assumes the learner is able to find the confidence interval with any $\delta$, which is equivalent to knowing the cumulative density function of the distribution and can be unrealistic for real human feedback data in practice. This upper bound result provides a more direct comparison with the preference-based approach and demonstrates how human bias can affect the suboptimality when the uncertainty noise can be coped with.

**Theorem 5.** Suppose Assumption 1 holds. For any fixed constant $0 < \delta < 1$, if one runs Algorithm 1 with

$$b_m = c_b \sqrt{\frac{V_{R,\sigma}^2 \log^q \frac{SA}{\delta}}{m}}$$

and an appropriately chosen universal constant $c_b$, as soon as $n > 8 \log \frac{2SA}{\delta} / \bar{d}$, where $\bar{d} := \min_{(s,a) \in \mathcal{X}} d(s,a)$ and $\mathcal{X} := \{(s,a) \in \mathcal{S} \times \mathcal{A} : d(s,a) > 0\}$, with probability $1 - \delta$, the suboptimality of the output policy $\widehat{\pi}_{\text{LCB}}$ satisfies

$$\text{SubOpt}(\widehat{\pi}_{\text{LCB}}) \le c_0 \sum_{(s,a) \in \mathcal{X}} d_\rho^{\pi^\star}(s,a) \cdot \bar{h}^{-1} \left( \sqrt{\frac{V_{R,\sigma}^2 \log^q \frac{SA}{\delta}}{n \cdot d(s,a)}} \right),$$

where $c_0$ is a constant that depends on $q$.

This theorem shows that even when the algorithm has full knowledge of the human uncertainty in the rating model, human bias can still influence the suboptimality of $\widehat{\pi}_{\text{LCB}}$ negatively. It demonstrates the effect of bias on the suboptimality is truly unavoidable when using human rating directly. This can be further illustrated with our example model (11) as follows, which shows the suboptimality still decays more slowly because of the quadratic human bias.

**Corollary 3.** Suppose Assumption 1 holds. For any fixed constant $0 < \delta < 1$, if one runs Algorithm 1 with

$$b_m = \sqrt{\frac{64\sigma^4 \log \frac{SA}{\delta}}{m} + \frac{8\sigma^2 \log \frac{SA}{\delta}}{m}},$$

as soon as $n > 8 \log \frac{2SA}{\delta} / \bar{d}$, where $\bar{d} := \min_{(s,a) \in \mathcal{X}} d(s,a)$ and $\mathcal{X} := \{(s,a) \in \mathcal{S} \times \mathcal{A} : d(s,a) > 0\}$, with probability $1 - \delta$, the suboptimality of the output policy $\widehat{\pi}_{\text{LCB}}$ satisfies

$$\text{SubOpt}(\widehat{\pi}_{\text{LCB}}) \le 2 \sum_{(s,a) \in \mathcal{X}} d_\rho^{\pi^\star}(s,a) \left( \frac{256\sigma^4 \log \frac{SA}{\delta}}{n \cdot d(s,a)} \right)^{1/4} + d_\rho^{\pi^\star}(s,a) \sqrt{\frac{32\sigma^2 \log \frac{SA}{\delta}}{n \cdot d(s,a)}}.$$

## D  SUPPORTING LEMMAS

**Lemma 1** (Hoeffding's inequality). Given $X_1, \cdots, X_n$ independent sub-gaussian random variables, each $X_i$ with variance proxy $\sigma_i^2$. It holds that

$$\mathbb{P}\left[ \sum_{i=1}^n (X_i - \mathbb{E}[X_i]) \ge t \right] \le \exp\left( -\frac{t^2}{2 \sum_{i=1}^n \sigma_i^2} \right).$$

**Lemma 2** (Bernstein's inequality). Given $X_1, \cdots, X_n$ independent sub-exponential random variables, each $X_i$ with parameters $(\tau_i^2, \alpha_i)$. Define

$$\tau_*^2 := \sum_{i=1}^n \tau_i^2 \quad \text{and} \quad \alpha_* := \max_i \alpha_i.$$

It holds that

$$\mathbb{P}\left[ \sum_{i=1}^n (X_i - \mathbb{E}[X_i]) \ge t \right] \le \begin{cases} \exp\left( -\frac{t^2}{2\tau_*^2} \right), & \text{if } 0 < t < \frac{\tau_*^2}{\alpha_*}; \\ \exp\left( -\frac{t}{2\alpha_*} \right), & \text{if } t \ge \frac{\tau_*^2}{\alpha_*}. \end{cases}$$

The following lemma is modified from Lemma B.1 in [66]. Recall $n_{(s,a)}$ denotes the number of samples in the offline dataset that visits $(s,a)$. This lemma is about the concentration of $n_{(s,a)}$, the number of samples in the offline dataset that visits $(s,a)$.

**Lemma 3.** Given a dataset $\mathcal{D}$ with $n$ i.i.d. samples $\mathcal{D} = \{(s_i, a_i)\}_{i=1}^n$ from a sampling distribution $d$, let $d(s,a)$ be the probability $(s,a)$ is sampled from the outcome space $\mathcal{S} \times \mathcal{A}$ and $n_{(s,a)}$ be the number of samples in $\mathcal{D}$ for $(s,a)$. For any $\delta \in (0,1)$, the event

$$d(s,a)\frac{n}{2} \le n_{(s,a)} \le d(s,a)\frac{3n}{2} \tag{13}$$

holds simultaneously for all $(s,a) \in \mathcal{S} \times \mathcal{X}$ with probability $1 - \delta$, as soon as $n > 8SA \log \frac{2SA}{\delta} / \bar{d}$, where $\bar{d} := \min_{(s,a) \in \mathcal{X}} d(s,a)$ and $\mathcal{X} := \{(s,a) \in \mathcal{S} \times \mathcal{A} : d(s,a) > 0\}$

*Proof.* First, consider the event $d(s,a)\frac{n}{2} > n_{(s,a)}$ for some fixed $(s,a) \in \mathcal{S} \times \mathcal{A}$.

Note that we can view $n_{(s,a)}$ as a sum of independent Bernoulli variables, i.e., $n_{(s,a)} = \sum_{i=1}^{n} \mathbb{1}\{(s_i, a_i) = (s,a)\}$, so $n_{(s,a)}$ follows a binomial distribution with parameters $(d(s,a), n)$. Recall the multiplicative Chernoff bound on a binomial random variable:

**Lemma 4** (Multiplicative Chernoff bound [10]). *Let $X$ be a Binomial random variable with parameters $(p, n)$. Denote its mean with $\mu$. For any $\epsilon \in (0, 1]$, we have*

$$\mathbb{P}\left[X < (1 - \epsilon)\mu\right] < e^{-\frac{\epsilon^2 \mu}{2}} \tag{14}$$

*and*

$$\mathbb{P}\left[X \geq (1 + \epsilon)\mu\right] < e^{-\frac{\epsilon^2 \mu}{3}}. \tag{15}$$

Take $\epsilon = \frac{1}{2}$. (14) suggests $d(s,a)\frac{n}{2} > n_{(s,a)}$ holds with probability at most $e^{-nd(s,a)/8}$. Similarly, taking $\epsilon = \frac{1}{2}$ in (15) suggests $n_{(s,a)} > d(s,a)\frac{3n}{2}$ holds with probability at most $e^{-nd(s,a)/12}$. Taking the union bound on these two events, we have $d(s,a)\frac{n}{2} > n_{(s,a)}$ or $n_{(s,a)} > d(s,a)\frac{3n}{2}$ with probability at most $2e^{-n\bar{d}/8}$ for this fixed $(s,a)$.

Then, we take a union bound over all $(s,a) \in \mathcal{S} \times \mathcal{A}$, which can give us the advertised result. $\qquad\square$

Moreover, since the uncertainty in our rating model in 3 can end up with a complex concentration, we need a lemma that can give the upper and lower tail bounds of Gaussian chaos.

**Lemma 5** (Corollary 1 in [25]). *Let $X$ be a zero-mean Gaussian random variable, and let $f : \mathbb{R} \to \mathbb{R}$ be a polynomial for degree $q \in \mathbb{Z}_{>0}$. Then*

$$\mathbb{P}\left[|f(X) - \mathbb{E}[f(X)]| \geq t\right] \geq c_q \exp\left(-\left(\frac{t^2}{c_1 \operatorname{Var}(f(X))}\right)^{1/q}\right) \tag{16}$$

*and*

$$\mathbb{P}\left[|f(X) - \mathbb{E}[f(X)]| \geq t\right] \leq C_q \exp\left(-\left(\frac{t^2}{C_1 \operatorname{Var}(f(X))}\right)^{1/q}\right), \tag{17}$$

*where $c_1, C_1 > 0$ are absolute constants and $c_q, C_q > 0$ are absolute constants depending on $q$.*

**Proposition 1.** *Let $\epsilon \sim \mathcal{N}(0, \sigma^2)$. $\epsilon|\epsilon|$ is a sub-exponential random variable with parameters $(4\sigma^4, 4\sigma^2)$.*

*Proof.* Let $X = Z^2$, where $Z \sim \mathcal{N}(0, 1)$. For $0 < \lambda < \frac{1}{2}$,

$$\mathbb{E}[e^{\lambda(X-1)/2}] = (1 - 2\lambda)^{-1/2}e^{-\lambda} = e^{-\frac{1}{2}\log(1-2\lambda) - \lambda} \geq e^{\lambda^2}$$

and by [52], for $0 < \lambda \leq \frac{1}{4}$,

$$\mathbb{E}[e^{\lambda(X-1)/2}] \leq e^{2\lambda^2}.$$

Thus, let $Y = (\sigma Z)^2$. For $0 < \lambda < \frac{1}{2\sigma^2}$,

$$\mathbb{E}[e^{\lambda(Y-\sigma^2)/2}] = e^{-\frac{1}{2}\log(1-2\lambda\sigma^2) - \lambda\sigma^2} \geq e^{\lambda^2\sigma^4}$$

and by [52], for $0 < \lambda \leq \frac{1}{4\sigma^2}$,

$$\mathbb{E}[e^{\lambda(Y-\sigma^2)/2}] \leq e^{2\lambda^2\sigma^4}.$$

$\qquad\square$

# E PROOFS FOR HUMAN RATING UNDER ASSUMPTION 1

## E.1 PROOF OF THEOREM 1

*Proof.* To prove this theorem, it suffices to construct a bandit instance with $\mathcal{S} = \{s\}$ and $\mathcal{A} = \{a_1, \cdots, a_A\}$, where $A = n^2$. Let the true reward function $r$ be defined such that two conditions are satisfied: (i) let $r(s, a_1) = r(s, a_i) + c(R, \sigma, q)\bar{h}^{-1}(V_{R,\sigma})$ for any $a_i$ such that $i = 2, \cdots, A$, where $c(R, \sigma, q)$ is a constant that might depend on $R, q, \sigma$ and makes sure that $c(R, \sigma, q)\bar{h}^{-1}(V_{R,\sigma}) = cR$ for some absolute constant $c > 0$ and $r \in [0, R]^{SA}$; (ii) let $\operatorname{Var}\left(h(r(s, a_i), \epsilon) - \bar{h}(r(s, a_i))\right) = c(q)V_{R,\sigma}^2$ for any $a_i$ such that $i = 2, \cdots, A$, where $c(q)$ is a constant depending on $q$. We can select

$c(R, \sigma, q)$ and $c(q)$ so that such a reward function $r$ exists. It can be observed that $a_1$ is the optimal action at $s$.

For the sampling distribution $d$, let $d(s, a_1) = \frac{1}{2}$ and $d(s, a) = \frac{1}{2(A-1)}$ for any $a_i$ such that $i = 2, \cdots, A$.

Consider the event that each of the suboptimal actions is observed less than once in the offline dataset, i.e., $n_{(s,a)} = 0$ or $n_{(s,a)} = 1$ for all $a_i$ such that $i = 2, \cdots, A$. For convenience, let us denote this event with $\mathcal{E}_1$. Conditioned on the total number of observations of suboptimal actions in the dataset, we can obtain the following via a simple counting argument

$$\mathbb{P}\left[\mathcal{E}_1 \;\middle|\; \sum_{i=2}^{A} n_{(s,a_i)}\right] = \frac{(A-1)(A-2)\cdots(A - \sum_{i=2}^{A} n_{(s,a_i)})}{(A-1)^{\sum_{i=2}^{A} n_{(s,a_i)}}} \geq \left(\frac{A-n}{A-1}\right)^n. \tag{18}$$

where the last inequality is because $(\frac{A-x}{A-1})^x$ is monotonically decreasing for $x \geq 1$ and the fact $n \geq \sum_{i=2}^{A} n_{(s,a_i)}$.

In addition, let us plug in $A = n^2$, and we can observe
$$\left(\frac{A-n}{A-1}\right)^n \geq \lim_{n \to \infty} \left(\frac{n^2-n}{n^2-1}\right)^n = \frac{1}{e}.$$
Combining the two inequalities above, we can arrive at $\mathbb{P}[\mathcal{E}_1 \mid \sum_{i=2}^{A} n_{(s,a_i)}] \geq \frac{1}{e}$. Since this bound holds for any value of $\sum_{i=2}^{A} n_{(s,a_i)}$, we have $\mathbb{P}[\mathcal{E}_1] \geq \frac{1}{e}$.

On the other hand, let us consider the event $n_{(s,a_1)} < 0.9n$ and denote it with $\mathcal{E}_2$. Note that this event is equivalent to the event $\sum_{i=2}^{A} n_{(s,a_i)} \geq 0.1n$. Since $n_{(s,a_1)}$ follows the binomial distribution with parameters $(n, \frac{1}{2})$, by (14) in Lemma 4, we have

$$\mathbb{P}\left[n_{(s,a_1)} < 0.9n\right] = 1 - \mathbb{P}\left[n_{(s,a_1)} > 0.9n\right] \geq 1 - e^{-8n/75} \geq 0.99517 \tag{19}$$
as long as $n \geq 50$.

Now, we are ready to give a lower bound for the probability of the event that a suboptimal arm is finally selected by Algorithm 1, i.e., there exists $a \neq a^\star, \widehat{r}(s, a) > \widehat{r}(s, a^\star)$.

To this end, let us define a new event $\mathcal{E} := \mathcal{E}_1 \cap \mathcal{E}_2$. The aforementioned probability can be decomposed as follows:

$$\mathbb{P}\left[\exists a \neq a^\star, \widehat{r}(s, a) > \widehat{r}(s, a^\star)\right]$$
$$= 1 - \mathbb{P}\left[\forall a \neq a^\star, \widehat{r}(s, a) \leq \widehat{r}(s, a^\star)\right]$$
$$= 1 - \left(\mathbb{P}\left[\mathcal{E} \wedge \forall a \neq a^\star, \widehat{r}(s, a) \leq \widehat{r}(s, a^\star)\right] + \mathbb{P}\left[\mathcal{E}^C \wedge \forall a \neq a^\star, \widehat{r}(s, a) \leq \widehat{r}(s, a^\star)\right]\right)$$
$$\geq 1 - \left(\mathbb{P}\left[\mathcal{E} \wedge \forall a \neq a^\star, \widehat{r}(s, a) \leq \widehat{r}(s, a^\star)\right] + \mathbb{P}\left[\mathcal{E}^C\right]\right).$$

Let us first focus on finding an upper bound for the probability of the intersection of $\mathcal{E}$ and the event that the optimal arm is selected by the algorithm correctly, i.e., for all $a \neq a^\star, \widehat{r}(s, a) \leq \widehat{r}(s, a^\star)$.

Consider a fixed suboptimal action $a \neq a^\star$ with only one observation in the dataset, i.e., $n_{(s,a)} = 1$. We have
$$\mathbb{P}\left[\widehat{r}(s, a) \leq \widehat{r}(s, a^\star)\right]$$
$$= \mathbb{P}\left[\left(h(r(s, a), \epsilon) - \bar{h}(r(s, a))\right) - \left(\widetilde{r}(s, a^\star) - \bar{h}(r(s, a^\star))\right)\right.$$
$$\left. \leq \bar{h}(r(s, a^\star)) - \bar{h}(r(s, a)) + b_1 - b_{n_{(s,a^\star)}}\right]$$
$$\leq \mathbb{P}\left[\left(h(r(s, a), \epsilon) - \bar{h}(r(s, a))\right) - \left(\widetilde{r}(s, a^\star) - \bar{h}(r(s, a^\star))\right) \leq \bar{h}(r(s, a^\star)) - \bar{h}(r(s, a)) + b_1\right]$$
$$\stackrel{(i)}{=} \mathbb{P}\left[\left(h(r(s, a), \epsilon) - \bar{h}(r(s, a))\right) + \left(\widetilde{r}(s, a^\star) - \bar{h}(r(s, a^\star))\right) \leq \bar{h}(r(s, a^\star)) - \bar{h}(r(s, a)) + b_1\right]$$
$$= 1 - \mathbb{P}\left[\left(h(r(s, a), \epsilon) - \bar{h}(r(s, a))\right) + \left(\widetilde{r}(s, a^\star) - \bar{h}(r(s, a^\star))\right)\right.$$

$$\geq \bar{h}(r(s,a^\star)) - \bar{h}(r(s,a)) + b_1 \Big]$$

$$\overset{(ii)}{\leq} 1 - c_q \exp\left(-\left(\frac{\left(\bar{h}(r(s,a^\star)) - \bar{h}(r(s,a)) + b_1\right)^2}{c_1\left(\mathrm{Var}\left(h(r(s,a),\epsilon) - \bar{h}(r(s,a))\right) + \mathrm{Var}\left(\widetilde{r}(s,a^\star) - \bar{h}(r(s,a^\star))\right)\right)}\right)^{1/q}\right)$$

$$\leq 1 - c_q \exp\left(-\left(\frac{\left(\bar{h}(r(s,a^\star)) - \bar{h}(r(s,a)) + b_1\right)^2}{c_1 \,\mathrm{Var}\left(h(r(s,a),\epsilon) - \bar{h}(r(s,a))\right)}\right)^{1/q}\right).$$

In the series of equalities and inequalities above, (i) is Condition 2 of $\mathcal{H}$ described in Section 3, which the uncertainty has symmetric concentration.

(ii) can be obtained by applying (16) in Lemma 5 to $\widetilde{r}(s,a^\star) - \bar{h}(r(s,a^\star))$, which is a sum of $n_{(s,a^\star)}$ human rating samples at $(s,a^\star)$ and one sample at $(s,a)$.

Recall that the pessimism penalty in Algorithm 1 is

$$b_n(s,a) = c_b \sqrt{\frac{\widetilde{V}_{R,\sigma}^2 \log \frac{SA}{\delta}}{n}}.$$

Bringing this and everything else into the inequality above, we have

$$\mathbb{P}\left[\widehat{r}(s,a) \leq \widehat{r}(s,a^\star)\right]$$

$$\leq 1 - c_q \exp\left(-\left(\frac{\left(\bar{h}(r(s,a^\star)) - \bar{h}(r(s,a)) + b_1\right)^2}{c_1 \,\mathrm{Var}\left(h(r(s,a),\epsilon) - \bar{h}(r(s,a))\right)}\right)^{1/q}\right)$$

$$\overset{(iii)}{\leq} 1 - c_q \exp\left(-\frac{\left(\bar{h}(r(s,a^\star)) - \bar{h}(r(s,a))\right)^{2/q} + b_1^{2/q}}{c_1^{2/q}\,\mathrm{Var}\left(h(r(s,a),\epsilon) - \bar{h}(r(s,a))\right)^{2/q}}\right)$$

$$\overset{(iv)}{\leq} 1 - c_q \exp\left(-\frac{\left(C_{h,2}\bar{h}\left(r(s,a^\star) - r(s,a)\right)\right)^{2/q} + b_1^{2/q}}{c_1^{2/q}\,\mathrm{Var}\left(h(r(s,a),\epsilon) - \bar{h}(r(s,a))\right)^{2/q}}\right)$$

$$= 1 - c_q \exp\left(-\frac{\left(C_{h,2}c(R,\sigma,q)V_{R,\sigma}\right)^{2/q} + b_1^{2/q}}{c_1^{2/q}\,\mathrm{Var}\left(h(r(s,a),\epsilon) - \bar{h}(r(s,a))\right)^{2/q}}\right)$$

$$= 1 - c_{q,1} \exp\left(-\frac{C_{h,2}^{2/q}(c(R,\sigma,q))^{2/q}V_{R,\sigma}^{2/q} + \left(c_b^2\widetilde{V}_{R,\sigma}^2 \log \frac{SA}{\delta}\right)^{1/q}}{c_1^{2/q}(c(q))^{2/q}V_{R,\sigma}^{2/q}}\right)$$

$$= 1 - c_{q,1} \exp\left(-\frac{C_{h,2}^{2/q}(c(R,\sigma,q))^{2/q}V_{R,\sigma}^{2/q} + \left(c_b^2 c_V V_{R,\sigma}^2 \log \frac{SA}{\delta}\right)^{1/q}}{c_1^{2/q}(c(q))^{2/q}V_{R,\sigma}^{2/q}}\right)$$

$$= 1 - c_{q,1}e^{-\left(\frac{C_{h,2}c(R,\sigma,q)}{c_1 c(q)}\right)^{2/q}} \exp\left(-\left(\frac{c_b\sqrt{c_V}}{c_1 c(q)}\right)^{2/q} \log^{1/q} \frac{n^2}{\delta}\right). \tag{20}$$

In the inequalities above, (iii) can be obtained by Jensen's inequality, because the function $x^{2/q}$ is concave for $x \geq 0$ when $q \geq 2$.

(iv) is obtained by Condition 3 of $\mathcal{H}$ described in Section 3.

We can further obtain the following once $n \geq c(q, c_b, c_V, \delta, \sigma, R)$:

$$\mathbb{P}\left[\mathcal{E} \wedge \forall a \neq a^\star, \widehat{r}(s,a) \leq \widehat{r}(s,a^\star)\right]$$

$$= \left( \mathbb{P}\left[ \widehat{r}(s, a_2) \leq \widehat{r}(s, a^\star) \right] \right)^{0.1n}$$

$$\leq \left( 1 - c_{q,1} e^{-\left( \frac{C_{h,2} c(R,\sigma,q)}{c_1 c(q)} \right)^{2/q}} \exp\left( -\left( \frac{c_b \sqrt{c_V}}{c_1 c(q)} \right)^{2/q} \log^{1/q} \frac{n^2}{\delta} \right) \right)^{0.1n}$$

$$\overset{(v)}{\leq} \left( 1 - c_{q,1} e^{-\left( \frac{C_{h,2} c(R,\sigma,q)}{c_1 c(q)} \right)^{2/q}} \left( \frac{\delta^{1/q}}{n^{2/q}} \right)^{\left( \frac{c_b \sqrt{c_V}}{c_1 c(q)} \right)^{2/q}} \right)^{0.1n} \tag{21}$$

$$\overset{(vi)}{<} 0.17. \tag{22}$$

Above, (v) becomes true once $n$ surpasses some threshold that depends on $q, c_b, c_V, \delta, c_1, C_{h,2}, \sigma, R$. For the case $q \geq 3$, the quantity in (21) decreases monotonically after a certain point that depends on $q, c_b, c_V, \delta, c_1, C_{h,2}, \sigma, R$ and its limit goes to $0$ as $n$ approaches infinity, so (vi) is true once $n \geq c(q, c_b, c_V, \delta, \sigma, R)$, where $c(q, c_b, c_V, \delta, \sigma, R)$ is a constant depending on $q, c_b, c_V, \delta, \sigma, R$. For the case $q = 2$, the quantity in (21) monotonically increases towards its limit, which is strictly less than 1, and we can choose $c(R, \sigma, q)$ and $c(q)$ appropriately so that (vi) holds.

On the other hand, through a union bound argument, we have

$$\mathbb{P}\left[ \mathcal{E}^C \right] \leq \mathbb{P}\left[ \mathcal{E}_1^C \right] + \mathbb{P}\left[ \mathcal{E}_2^C \right] = \left( 1 - \frac{1}{e} \right) - (1 - 0.99517) < 0.63. \tag{23}$$

Overall, combining (22) and (23), we can obtain a lower bound on the probability that Algorithm 1 fails to identify the optimal policy:

$$\mathbb{P}\left[ \exists a \neq a^\star, \widehat{r}(s, a) > \widehat{r}(s, a^\star) \right]$$
$$\geq 1 - \left( \mathbb{P}\left[ \mathcal{E} \wedge \forall a \neq a^\star, \widehat{r}(s, a) \leq \widehat{r}(s, a^\star) \right] + \mathbb{P}\left[ \mathcal{E}^C \right] \right)$$
$$> 1 - 0.17 - 0.63$$
$$= 0.2.$$

Finally, using this probability lower bound, we arrive at the desired lower bound on the excepted suboptimality:

$$\mathbb{E}_{\mathcal{D}}[\text{SubOpt}(\widehat{\pi}_{\text{LCB}})] = \mathbb{P}\left[ \exists a \neq a^\star, \widehat{r}(s, a) > \widehat{r}(s, a^\star) \right] \cdot (r(s, a^\star) - r(s, a))$$
$$= 0.2 c(R, \sigma, q) \bar{h}^{-1} \left( V_{R,\sigma} \right)$$
$$= c_0 R.$$

$\square$

### E.2 PROOF OF COROLLARY 1

*Proof.* To prove this corollary, we can construct a bandit instance with the same state space $\mathcal{S}$ and action space $\mathcal{A}$ as in the proof for Theorem 1. Recall the specific human rating function for this corollary is $h(r, \epsilon) = r^2 + r^2 \epsilon |\epsilon|$ with $\bar{h}(r) = r^2$. The variance of $h(r, \epsilon)$ is $3 r^4 \sigma^4$. Let the true reward function $r$ be defined such that $r(s, a_1) = \frac{1}{2} + \frac{1}{4 \cdot 3^{1/4} \sigma} (3 r^4 \sigma^4)^{1/4} = \frac{3}{4}$ and $r(s, a_2) = \frac{1}{2}$. It can be checked that this reward function satisfies the two conditions in the proof for Theorem 1. The remaining of the proof is similar to the proof for Theorem 1. We can conclude that Algorithm 1 suffers a suboptimality of $\frac{1}{4}$ with constant probability depending on $q, c_b, c_V, \delta, \sigma$. $\square$

## F PROOFS FOR HUMAN RATING UNDER ASSUMPTION 2

### F.1 PROOF OF THEOREM 2

*Proof.* To prove this theorem, it suffices to construct a bandit instance with $\mathcal{S} = \{s\}$ and $\mathcal{A} = \{a_1, a_2\}$. Let the true reward function $r$ be defined such that two conditions are satisfied: (i) let $r(s, a_1) = r(s, a_2) + \bar{h}^{-1} \left( \sqrt{\frac{V_{R,\sigma}^2}{n}} \right)$. Note that when $n$ is sufficiently large, i.e., when $n \geq c(R, \sigma, q)$, we can make sure $r \in [0, R]^{SA}$. For (ii), we let $\text{Var}\left( h(r(s, a_1), \epsilon) - \bar{h}(r(s, a_i)) \right) = c_1(q) V_{R,\sigma}^2$ and $\text{Var}\left( h(r(s, a_2), \epsilon) - \bar{h}(r(s, a_i)) \right) = c_2(q) V_{R,\sigma}^2$, where $c_1(q)$ and $c_2(q)$ are constants

depending on $q$. We can select $c(R, \sigma, q)$, $c_1(q)$ and $c_2(q)$ so that such a reward function $r$ exists. It can be observed that $a_1$ is the optimal action at $s$. Also recall the sampling distribution $d$ is uniform, i.e., $d(s, a) = \frac{1}{SA}$ for any $(s, a) \in \mathcal{S} \times \mathcal{A}$.

Let us consider the regime when $n > 8SA \log(\frac{SA}{0.1})/\bar{d}$, where $\bar{d} := \min_{s,a}\{d(s, a) : d(s, a) > 0\}$. In our setting, this is equivalent to $n \geq 120 > 32 \log(40)$. By Lemma 3, for all $(s, a) \in \mathcal{S} \times \mathcal{A}$ simultaneously,

$$\frac{1}{2}n \cdot d(s, a) \leq n_{(s,a)} \leq \frac{3}{2}n \cdot d(s, a) \tag{24}$$

with probability at least 0.9.

In the event of (24), we have $cn_{(s,a_1)} = n_{(s,a_2)}$ with $\frac{1}{3} \leq c \leq 3$. Since we can design the reward function adversarially so that $a_1$ is the arm with more samples in the offline dataset, we can assume $1 \leq c_s \leq 3$ without loss of generality.

Similar to the proof of Theorem 1, we want to give a lower bound for the probability of the event that a suboptimal arm is finally selected by Algorithm 1, i.e., $\hat{r}(s, a_2) > \hat{r}(s, a_1)$. We can rewrite this probability as follows:

$$\mathbb{P}\left[\hat{r}(s, a_2) > \hat{r}(s, a_1)\right]$$

$$= \mathbb{P}\left[\tilde{r}(s, a_2) - b_{n_{(s,a_2)}} > \tilde{r}(s, a_1) - b_{n_{(s,a_1)}}\right]$$

$$= \mathbb{P}\left[\tilde{r}(s, a_2) - \tilde{r}(s, a_1) > b_{n_{(s,a_2)}} - b_{n_{(s,a_1)}}\right]$$

$$= \mathbb{P}\left[\left(\tilde{r}(s, a_2) - \bar{h}(r(s, a_2))\right) - \left(\tilde{r}(s, a_1) - \bar{h}(r(s, a_1))\right) \right.$$

$$\left. > \bar{h}(r(s, a_1)) - \bar{h}(r(s, a_2)) + b_{n_{(s,a_2)}} - b_{n_{(s,a_1)}}\right]$$

$$= \mathbb{P}\left[\left(\tilde{r}(s, a_2) - \bar{h}(r(s, a_2))\right) + \left(\tilde{r}(s, a_1) - \bar{h}(r(s, a_1))\right) \right.$$

$$\left. > \bar{h}(r(s, a_1)) - \bar{h}(r(s, a_2)) + b_{n_{(s,a_2)}} - b_{n_{(s,a_1)}}\right],$$

where the last step is due to Condition 2 of $\mathcal{H}$ described in Section 3, which the uncertainty has symmetric concentration.

We can invoke (16) in Lemma 5 on the quantity $\left(\tilde{r}(s, a_2) - \bar{h}(r(s, a_2))\right) - \left(\tilde{r}(s, a_1) - \bar{h}(r(s, a_1))\right)$ above, which is a sum of $n_{(s,a_1)}$ human rating samples at $(s, a_1)$ and $n_{(s,a_2)}$ human rating samples at $(s, a_2)$. This can give us:

$$\mathbb{P}\left[\left(\tilde{r}(s, a_2) - \bar{h}(r(s, a_2))\right) + \left(\tilde{r}(s, a_1) - \bar{h}(r(s, a_1))\right) \right.$$

$$\left. > \bar{h}(r(s, a_1)) - \bar{h}(r(s, a_2)) + b_{n_{(s,a_2)}} - b_{n_{(s,a_1)}}\right]$$

$$\geq c_q \exp\left(-\left(\frac{\left(\bar{h}(r(s, a_1)) - \bar{h}(r(s, a_2))\right)^2}{c_1(\frac{c_1(q)}{n_{(s,a_1)}} + \frac{c_2(q)}{n_{(s,a_2)}})V_{R,\sigma}^2}\right)^{1/q}\right)$$

$$\geq c_q \exp\left(-\left(\frac{\left(C_{h,2}\bar{h}(r(s, a_1) - r(s, a_2))\right)^2}{c_1(\frac{c_1(q)}{n_{(s,a_1)}} + \frac{c_2(q)}{n_{(s,a_2)}})V_{R,\sigma}^2}\right)^{1/q}\right)$$

$$= c_q \exp\left(-\left(\frac{C_{h,2}^2 V_{R,\sigma}^2}{nc_1(\frac{c_1(q)}{n_{(s,a_1)}} + \frac{c_2(q)}{n_{(s,a_2)}})V_{R,\sigma}^2}\right)^{1/q}\right)$$

$$\geq c_q \exp\left(-\left(\frac{C_{h,2}^2 V_{R,\sigma}^2}{nc_1(\frac{c_1(q)}{\frac{3}{2}n \cdot d(s,a_1)} + \frac{c_2(q)}{\frac{3}{2}n \cdot d(s,a_2)})V_{R,\sigma}^2}\right)^{1/q}\right)$$

$$\geq c_q \exp\left(-\left(\frac{3C_{h,2}^2}{4c_1(c_1(q) + c_2(q))}\right)^{1/q}\right)$$

$$=: c_0.$$

Overall, Algorithm 1 is guaranteed to incur an expected suboptimality at least $c_0 \bar{h}^{-1}\left(\sqrt{\frac{V_{R,\sigma}^2}{n}}\right)$ as soon as $n \geq \max\{c(R, \sigma, q), 120\}$. $\qquad\square$

### F.2 Proof of Corollary 2

*Proof.* To prove this corollary, we can construct a bandit instance with the same state space $\mathcal{S}$ and action space $\mathcal{A}$ as in the proof for Theorem 2. Recall the specific human rating function for this corollary is $h(r, \epsilon) = r^2 + r^2\epsilon|\epsilon|$ with $\bar{h}(r) = r^2$. The variance of $h(r, \epsilon)$ is $3r^4\sigma^4$. Let the true reward function $r$ be defined such that $r(s, a_1) = \frac{1}{2} + \sigma\left(\frac{1}{n}\right)^{1/4}$ and $r(s, a_2) = \frac{1}{2}$. It can be checked that this reward function satisfies the two conditions in the proof for Theorem 1. The remaining of the proof is similar to the proof for Theorem 1. We can conclude that Algorithm 1 suffers a suboptimality of $\sigma\left(\frac{1}{n}\right)^{1/4}$ with constant probability depending on $q$ as soon as $n \geq \max\{48\sigma^4, 60\}$. $\qquad\square$

## G Proofs for human rating upper bounds

### G.1 Proof of Theorem 5

*Proof.* In this proof, let us denote the output of Algorithm 1 $\widehat{\pi}_{\mathrm{LCB}}$ with $\widehat{\pi}$ for short. By the definition in (1), we can decompose the suboptimality of the output $\widehat{\pi}$ of Algorithm 1 as follows:

$$\mathrm{SubOpt}(\widehat{\pi})$$
$$= \mathbb{E}_{s\sim\rho}[r(s, \pi^\star(s)) - r(s, \widehat{\pi}(s))]$$
$$= \mathbb{E}_{s\sim\rho}[(\bar{h}^{-1} \circ \bar{h})(r(s, \pi^\star(s))) - (\bar{h}^{-1} \circ \bar{h})(r(s, \widehat{\pi}(s)))]$$
$$\overset{(i)}{\leq} C_{h,1}\mathbb{E}_{s\sim\rho}\left[\bar{h}^{-1}\left(\bar{h}(r(s, \pi^\star(s))) - \bar{h}(r(s, \widehat{\pi}(s)))\right)\right]$$
$$= C_{h,1}\mathbb{E}_{s\sim\rho}\left[\bar{h}^{-1}\left(\bar{h}(r(s, \pi^\star(s))) - \widehat{r}(s, \pi^\star(s)) + \widehat{r}(s, \pi^\star(s)) - \widehat{r}(s, \widehat{\pi}(s)) + \widehat{r}(s, \widehat{\pi}(s)) - \bar{h}(r(s, \widehat{\pi}(s)))\right)\right]$$
$$\overset{(ii)}{\leq} C_{h,1}\mathbb{E}_{s\sim\rho}\left[\bar{h}^{-1}\left(\bar{h}(r(s, \pi^\star(s))) - \widehat{r}(s, \pi^\star(s)) + \widehat{r}(s, \widehat{\pi}(s)) - \bar{h}(r(s, \widehat{\pi}(s)))\right)\right]$$
$$\overset{(iii)}{\leq} C_{h,1}\mathbb{E}_{s\sim\rho}\left[\bar{h}^{-1}\left(\bar{h}(r(s, \pi^\star(s))) - \widehat{r}(s, \pi^\star(s))\right)\right]$$
$$\overset{(iv)}{\leq} C_{h,1}\mathbb{E}_{s\sim\rho}\left[\bar{h}^{-1}\left(\left|\bar{h}(r(s, \pi^\star(s))) - \widetilde{r}(s, \pi^\star(s))\right| + b_{n_{(s,\pi^\star(s))}}\right)\right]$$
$$\overset{(v)}{\leq} C_{h,1}\mathbb{E}_{(s,a)\sim d_\rho^{\pi^\star}}\left[\bar{h}^{-1}\left(C'\sqrt{\frac{V_{R,\sigma}^2 \log^q \frac{SA}{\delta}}{n_{(s,a)}}}\right)\right]$$
$$\overset{(vi)}{\leq} C_{h,1}\mathbb{E}_{(s,a)\sim d_\rho^{\pi^\star}}\left[\bar{h}^{-1}\left(C\sqrt{\frac{V_{R,\sigma}^2 \log^q \frac{SA}{\delta}}{n \cdot d(s,a)}}\right)\right]$$
$$\overset{(vii)}{\leq} c_0\mathbb{E}_{(s,a)\sim d_\rho^{\pi^\star}}\left[\bar{h}^{-1}\left(\sqrt{\frac{V_{R,\sigma}^2 \log^q \frac{SA}{\delta}}{n \cdot d(s,a)}}\right)\right]$$
$$\leq c_0 \sum_{(s,a)\in\mathcal{X}} d_\rho^{\pi^\star}(s,a) \cdot \bar{h}^{-1}\left(\sqrt{\frac{V_{R,\sigma}^2 \log^q \frac{SA}{\delta}}{n \cdot d(s,a)}}\right). \tag{25}$$

In the series of equalities and inequalities above, (i) is due to Condition 3 of $\mathcal{H}$ described in Section 3.

(ii) can be obtained because of the fact that in Algorithm 1, $\widehat{\pi}(s) = \arg\max_a \widehat{r}(s,a)$, so for every $s \in \mathcal{S}$, $\widehat{r}(s,\widehat{\pi}(s)) \geq \widehat{r}(s,\pi^\star(s))$.

(iii) can be obtained because the pessimistic penalty $b_n$ in Theorem 5 guarantees $\widehat{r}(s,\widehat{\pi}(s)) \leq \bar{h}(r(s,\widehat{\pi}(s)))$ with probability $1 - \delta$.

To show this, we can focus on bounding the probability of the event

$$\widetilde{r}(s,a) - \mathbb{E}[\widetilde{r}(s,a)] \leq b_{n_{(s,a)}}$$

for all $(s,a) \in \mathcal{S} \times \mathcal{A}$ simultaneously. For simplicity in the remaining part of this proof, let us denote this event with $\mathcal{E}$. This involves an understanding of the concentration of the rating observation $h(r(s,a), \epsilon)$ under (3).

By Lemma 5 and a union bound argument, for all $(s,a) \in \mathcal{S} \times \mathcal{A}$ simultaneously, we have

$$\mathbb{P}\left[|\widetilde{r}(s,a) - \mathbb{E}[\widetilde{r}(s,a)]| \geq t\right] \leq SA \cdot C_q \exp\left(-\left(\frac{t^2}{C_1 V_{R,\sigma}^2}\right)^{1/q}\right).$$

We can solve for $t$ in the inequality above. Given $b_n$ in Theorem 5 with a sufficiently large $c_b$, this gives

$$|\widetilde{r}(s,a) - \mathbb{E}[\widetilde{r}(s,a)]| \leq c(q)\sqrt{\frac{V_{R,\sigma}^2 \log^q \frac{SA}{\delta}}{n_{(s,a)}}} \leq b_{n_{(s,a)}} \tag{26}$$

for all $(s,a) \in \mathcal{S} \times \mathcal{A}$ simultaneously with probability at least $1 - \delta$. Here, $c(q)$ is an absolute constant depending on $q$.

Note that $\bar{h}(r(s,a)) = \mathbb{E}[\widetilde{r}(s,a)]$. Thus, we can establish (iii) on the event $\mathcal{E}$, which has probability at least $1 - \delta$.

(iv) is obtained by the monotoncity of $\bar{h}^{-1}(\cdot)$. Since $\bar{h}(r(s,a)) - \widehat{r}(s,a) \leq |\bar{h}(r(s,a)) - \widetilde{r}(s,a)| + b_{n_{(s,a)}}$, $\bar{h}^{-1}(\bar{h}(r(s,a)) - \widehat{r}(s,a)) \leq \bar{h}^{-1}(|\bar{h}(r(s,a)) - \widetilde{r}(s,a)| + b_{n_{(s,a)}})$, for any $(s,a) \in \mathcal{S} \times \mathcal{A}$.

(v) can be obtained by finding an upper bound for $|\bar{h}(r(s,\pi^\star(s))) - \widetilde{r}(s,\pi^\star(s))|$, which follows from (26). That is, on the event $\mathcal{E}$, we have

$$|\bar{h}(r(s,\pi^\star(s))) - \widetilde{r}(s,\pi^\star(s))| \leq c(q)\sqrt{\frac{V_{R,\sigma}^2 \log^q \frac{SA}{\delta}}{n_{(s,\pi^\star(s))}}}$$

for some absolute constant $c(q)$ depending on $q$. Then, the addition of the bound above and $b_{n_{(s,\pi^\star(s))}}$ gives (v).

(vi) can be obtained from an invocation of Lemma 3, which guarantees $n_{(s,a)} = cd(s,a)n$ for some constant $\frac{1}{2} \leq c \leq \frac{3}{2}$ for all $(s,a) \in \mathcal{S} \times \mathcal{A}$ simultaneously, with probability at least $1 - \delta$. We can denote this event with $\mathcal{E}'$.

Finally, (vii) is obtained because we can pull the constant factor inside $\bar{h}^{-1}$ out due to Condition 3 of $\mathcal{H}$ described in Section 3.

The advertised bound can be obtained after taking a union bound on the probability of the complement of $\mathcal{E}$ and the complement of $\mathcal{E}'$, which only changes (25) by a constant factor. $\qquad\square$

### G.2 Proof of Corollary 3

*Proof.* In this proof, let us denote the output of Algorithm 1 $\widehat{\pi}_{\text{LCB}}$ with $\widehat{\pi}$ for short. By the definition in (1), we can decompose the suboptimality of the output $\widehat{\pi}$ of Algorithm 1 as follows:

$$\begin{aligned}
&\text{SubOpt}(\widehat{\pi}) \\
&= \mathbb{E}_{s\sim\rho}[r(s,\pi^\star(s)) - r(s,\widehat{\pi}(s))] \\
&= \mathbb{E}_{s\sim\rho}\left[\sqrt{r^2(s,\pi^\star(s))} - \sqrt{r^2(s,\widehat{\pi}(s))}\right] \\
&\overset{(i)}{\leq} \mathbb{E}_{s\sim\rho}\left[\sqrt{r^2(s,\pi^\star(s)) - r^2(s,\widehat{\pi}(s))}\right] \\
&= \mathbb{E}_{s\sim\rho}\left[\sqrt{r^2(s,\pi^\star(s)) - \widehat{r}(s,\pi^\star(s)) + \widehat{r}(s,\pi^\star(s)) - \widehat{r}(s,\widehat{\pi}(s)) + \widehat{r}(s,\widehat{\pi}(s)) - r^2(s,\widehat{\pi}(s))}\right]
\end{aligned}$$

$$\overset{\text{(ii)}}{\leq} \mathbb{E}_{s\sim\rho}\left[\sqrt{r^2(s,\pi^\star(s)) - \widehat{r}(s,\pi^\star(s)) + \widehat{r}(s,\widehat{\pi}(s)) - r^2(s,\widehat{\pi}(s))}\right]$$

$$\overset{\text{(iii)}}{\leq} \mathbb{E}_{s\sim\rho}\left[\sqrt{r^2(s,\pi^\star(s)) - \widehat{r}(s,\pi^\star(s))}\right]$$

$$\overset{\text{(iv)}}{\leq} \mathbb{E}_{s\sim\rho}\left[\sqrt{\left|\bar{h}(r(s,\pi^\star(s))) - \widetilde{r}(s,\pi^\star(s))\right| + b_{n_{(s,\pi^\star(s))}}}\right]$$

$$\overset{\text{(v)}}{\leq} \mathbb{E}_{(s,a)\sim d_\rho^{\pi^\star}}\left[\left(2\sqrt{\frac{64\sigma^4\log\frac{SA}{\delta}}{n_{(s,a)}} + \frac{16\sigma^2\log\frac{SA}{\delta}}{n_{(s,a)}}}\right)^{1/2}\right]$$

$$\leq 2\sum_{(s,a)\in\mathcal{X}} d_\rho^{\pi^\star}(s,a)\left(\frac{256\sigma^4\log\frac{SA}{\delta}}{n\cdot d(s,a)}\right)^{1/4} + d_\rho^{\pi^\star}(s,a)\sqrt{\frac{32\sigma^2\log\frac{SA}{\delta}}{n\cdot d(s,a)}}. \tag{27}$$

In the series of equalities and inequalities above, (i) is due to Condition 3 of $\mathcal{H}$ described in Section 3, which can be proved for $\bar{h}(\cdot) = (\cdot)^2$ with Jensen's inequality.

(ii) can be obtained because of the fact that in Algorithm 1, $\widehat{\pi}(s) = \arg\max_a \widehat{r}(s,a)$, so for every $s\in\mathcal{S}, \widehat{r}(s,\widehat{\pi}(s)) \geq \widehat{r}(s,\pi^\star(s))$.

(iii) can be obtained because the pessimistic penalty $b_n$ in Corollary 3 guarantees $\widehat{r}(s,\widehat{\pi}(s)) \leq r^2(s,\widehat{\pi}(s))$ with probability $1-\delta/2$.

To show this, we can focus on bounding the probability of the event

$$\widetilde{r}(s,a) - \mathbb{E}[\widetilde{r}(s,a)] \leq b_{n_{(s,a)}}$$

for all $(s,a)\in\mathcal{S}\times\mathcal{A}$ simultaneously. For simplicity in the remaining part of this proof, let us denote this event with $\mathcal{E}$. This involves an understanding of the concentration of the rating observation $h(r(s,a),\epsilon)$ under (3).

Note that $\epsilon|\epsilon|$ is a sub-exponential random variable with parameters $(4\sigma^4, 4\sigma^2)$ by Proposition 1.

Now, by Lemma 2 and a union bound argument, for all $(s,a)\in\mathcal{S}\times\mathcal{A}$ simultaneously, we have

$$|\widetilde{r}(s,a) - \mathbb{E}[\widetilde{r}(s,a)]| \leq \sqrt{\frac{64\sigma^4\log\frac{SA}{\delta}}{n_{(s,a)}}} + \frac{8\sigma^2\log\frac{SA}{\delta}}{n_{(s,a)}} \leq b_{n_{(s,a)}}$$

with probability at least $1-\delta/2$.

Note that $\bar{h}(r(s,a)) = \mathbb{E}[\widetilde{r}(s,a)]$. Thus, we can establish (iii) on the event $\mathcal{E}$, which has probability at least $1-\delta/2$.

(iv) is obtained by the monotoncity of $\bar{h}^{-1}(\cdot) = \sqrt{\cdot}$. Since $\bar{h}(r(s,a)) - \widehat{r}(s,a) \leq |\bar{h}(r(s,a)) - \widetilde{r}(s,a)| + b_{n_{(s,a)}}, \bar{h}^{-1}(\bar{h}(r(s,a)) - \widehat{r}(s,a)) \leq \bar{h}^{-1}(|\bar{h}(r(s,a)) - \widetilde{r}(s,a)| + b_{n_{(s,a)}})$, for any $(s,a)\in\mathcal{S}\times\mathcal{A}$.

(v) can be obtained by finding an upper bound for $|\bar{h}(r(s,\pi^\star(s))) - \widetilde{r}(s,\pi^\star(s))|$, which follows from (26). That is, on the event $\mathcal{E}$, we have

$$|\bar{h}(r(s,\pi^\star(s))) - \widetilde{r}(s,\pi^\star(s))| \leq \sqrt{\frac{64\sigma^4\log\frac{SA}{\delta}}{n_{(s,a)}}} + \frac{8\sigma^2\log\frac{SA}{\delta}}{n_{(s,\pi^\star(s))}}.$$

Then, the addition of the bound above and $b_{n_{(s,\pi^\star(s))}}$ gives (v).

(vi) can be obtained from an invocation of Lemma 3, which guarantees $n_{(s,a)} = cd(s,a)n$ for some constant $\frac{1}{2}\leq c\leq\frac{3}{2}$ for all $(s,a)\in\mathcal{S}\times\mathcal{A}$ simultaneously, with probability at least $1-\delta/2$. We can denote this event with $\mathcal{E}'$.

The advertised bound can be obtained after taking a union bound on the probability of the complement of $\mathcal{E}$ and the complement of $\mathcal{E}'$, which changes (27) by a factor of 2. $\qquad\square$

## H  PROOF OF THEOREM 3

[77] considers the setting in which the reward function $r_{\theta^\star}(s,a)$ is parameterized by $\theta^\star\in\mathbb{R}^d$ in a linear fashion, i.e., $r_{\theta^\star}(s,a) = (\theta^\star)^\top\phi(s,a)$. Here, $\phi(s,a)$ is a known feature in $\mathbb{R}^d$. The

tabular setting we consider is a special case of the linear setting. To write the tabular setting in this notation, we can let $\phi(s, a) = e_{(s,a)}$, where $e_{(s,a)} \in \mathbb{R}^{SA}$ is the canonical basis vector for the $(s, a)$-th dimension, and let $\theta^\star \in \mathbb{R}^{SA}$ be a shifted version of the tabular reward function/vector $r \in [0, R]^{SA}$, i.e., $\theta^\star(s, a) = r(s, a) - \frac{1}{SA} \sum_{(s,a)} r(s, a) \in [-R, R]^{SA}$. This ensures $\mathbf{1}^\top \theta^\star = 0$, which is required by [77] for the identifiability of $\theta^\star$. As [77] shows, Algorithm 2 converges to $\theta^\star$.

In the analysis below, we omit the subscript of $\widehat{\pi}_{\mathrm{PMLE}}$ and write the output policy of Algorithm 2 as $\widehat{\pi}$ for simplicity. We also denote the shifted true reward with $\bar{r}(s, a) := r(s, a) - \frac{1}{SA} \sum_{(s,a)} r(s, a)$ for any $s \in \mathcal{S}$ and $a \in \mathcal{A}$ and let $\widehat{r}$ be the one from Line 6 of Algorithm 2. The following analysis largely overlaps with Theorem 3.2 in [77], but we handle the distributional mismatch term with more care and provide a characterization specific to the tabular setting.

The suboptimality of Algorithm 2 can be decomposed as follows:

$$
\begin{aligned}
&\mathrm{SubOpt}(\widehat{\pi}_{\mathrm{PMLE}}) \\
&= \mathbb{E}_{s \sim \rho}[r(s, \pi^\star(s)) - r(s, \widehat{\pi}(s))] \\
&= \mathbb{E}_{s \sim \rho}[\bar{r}(s, \pi^\star(s)) - \bar{r}(s, \widehat{\pi}(s))] \\
&= \mathbb{E}_{s \sim \rho}\Big[\bar{r}(s, \pi^\star(s)) - \widehat{r}(s, \pi^\star(s)) + \widehat{r}(s, \pi^\star(s)) - \widehat{r}(s, \widehat{\pi}(s)) + \widehat{r}(s, \widehat{\pi}(s)) - \bar{r}(s, \widehat{\pi}(s))\Big] \\
&\overset{\mathrm{(i)}}{\leq} \mathbb{E}_{s \sim \rho}\Big[\bar{r}(s, \pi^\star(s)) - \widehat{r}(s, \pi^\star(s)) + \widehat{r}(s, \widehat{\pi}(s)) - \bar{r}(s, \widehat{\pi}(s))\Big] \\
&\overset{\mathrm{(ii)}}{\leq} \mathbb{E}_{s \sim \rho}\Big[\bar{r}(s, \pi^\star(s)) - \widehat{r}(s, \pi^\star(s))\Big] \\
&\overset{\mathrm{(iii)}}{\leq} \sqrt{\sup_{v: \|v\|_\infty \leq 2R, \mathbf{1}^\top v = 0} \frac{\Big(\sum_{(s,a)} d_\rho^{\pi^\star}(s, a) v(s, a)\Big)^2}{\sum_{(s, a^0, a^1)} d(s, a^0, a^1)\big(v(s, a^0) - v(s, a^1)\big)^2}} \, \|\bar{r} - \widehat{r}\|_\Sigma \\
&\overset{\mathrm{(iv)}}{=} \sqrt{\sup_{v: \|v\|_\infty \leq 1, \mathbf{1}^\top v = 0} \frac{\Big(\sum_{(s,a)} d_\rho^{\pi^\star}(s, a) v(s, a)\Big)^2}{\sum_{(s, a^0, a^1)} d(s, a^0, a^1)\big(v(s, a^0) - v(s, a^1)\big)^2}} \, \|\bar{r} - \widehat{r}\|_\Sigma \\
&\leq C^\dagger \left(\|\bar{r} - \widetilde{r}\|_\Sigma + \|\widetilde{r} - \widehat{r}\|_\Sigma\right) \\
&\overset{\mathrm{(v)}}{\leq} 2C^\dagger \left(CR\sqrt{\frac{SA + \log \frac{1}{\delta}}{\gamma^2 n}} + cR\sqrt{\frac{S^2 A^2 \log \frac{n}{\delta}}{n}}\right) \\
&\leq c_0 C^\dagger R \left(\sqrt{\frac{SA + \log \frac{1}{\delta}}{\gamma^2 n}} + \sqrt{\frac{S^2 A^2 \log \frac{n}{\delta}}{n}}\right).
\end{aligned}
\tag{28}
$$

In the analysis above, (i) can be obtained because of Line 6 in Algorithm 2, $\widehat{\pi}(s) = \arg\max_\pi \mathbb{E}_{s \sim \rho}[\widehat{r}(s, \pi(s))]$, so $\mathbb{E}_{s \sim \rho}[\widehat{r}(s, \widehat{\pi}(s))] \geq \mathbb{E}_{s \sim \rho}[\widehat{r}(s, \pi^\star(s))]$.

(ii) is because $\widehat{r}$ is selected pessimistically from $\mathcal{F}_{\mathrm{CR}}(\widetilde{r})$ in Line 6 of Algorithm 2 and $\bar{r} \in \mathcal{F}_{\mathrm{CR}}(\widetilde{r})$ with probability at least $1 - \delta/2$. This is guaranteed by Lemma 3.1 of [77], which proves with probability at least $1 - \delta$,

$$
\|\widetilde{r} - \bar{r}\|_{\widehat{\Sigma}} \leq CR\sqrt{\frac{SA + \log \frac{1}{\delta}}{\gamma^2 n}},
\tag{29}
$$

where $C > 0$ is an absolute constant. Hence, we have $\mathbb{E}_{s \sim \rho}[\widehat{r}(s, \widehat{\pi}(s))] \leq \mathbb{E}_{s \sim \rho}[\bar{r}(s, \widehat{\pi}(s))]$.

In (iii), we change the measure by invoking the following lemma:

**Lemma 6** (modified from Lemma 1 in [19]). Given a function class $\mathcal{F}$ that contains functions mapping from $\mathcal{X}$ to $\mathbb{R}$ and a probability distribution $p_1$ supported on $\mathcal{X}$ and a joint quasiprobability probability distribution $p_2$ supported on $\mathcal{X} \times \mathcal{X}$, for any $g \in \mathcal{F}$,

$$
\mathbb{E}_{x \sim p_1}[g(x)] \leq \sqrt{\mathbb{E}_{(x,x') \sim p_2}[g(x)g(x')] \sup_{g \in \mathcal{F}} \frac{\mathbb{E}_{p_1}[g(x)]^2}{\mathbb{E}_{p_2}[g(x)g(x')]}}.
$$

Moreover, note that $\Sigma$ is a scaled Laplacian matrix. Since $\bar{r}, \widehat{r} \in \mathcal{F}$ ($\mathcal{F}$ is defined in Algorithm 2) and $v^\top \Sigma v = \sum_{i,j} \Sigma_{i,j} v_i v_j$ for any vector $v$, this allows us to write (the denominator of) the distributional mismatch term in the form of $C^\dagger$ defined in (8).

(iv) can be obtained because the distributional mismatch term $C^\dagger$ is constant with respect to any nonzero scaling of $v$.

(v) can be obtained by invoking the following lemma, which provides an upper bound on the difference between the ground truth $\bar{r}$ and the empirical estimation $\widetilde{r}$ with respect to $\Sigma$, the covariance matrix for the population sampling distribution.

**Lemma 7.** With probability at least $1 - \delta$, Algorithm 2 satisfies

$$\|\widetilde{r} - \bar{r}\|_\Sigma \leq CR\sqrt{\frac{SA + \log\frac{1}{\delta}}{\gamma^2 n}} + cR\sqrt{\frac{S^2 A^2 \log\frac{n}{\delta}}{n}}, \tag{30}$$

where $\gamma = \frac{1}{2 + \exp(R\sqrt{SA}) + \exp(-R\sqrt{SA})}$ and $C, c > 0$ are absolute constants.

The proof of Lemma 7 is deferred to Section H.1.

Finally, we conclude with (28), which leads to the advertised result.

### H.1 PROOF OF LEMMA 7

We can start with

$$\|\widetilde{r} - \bar{r}\|_\Sigma = \sqrt{\|\widetilde{r} - \bar{r}\|_\Sigma^2}$$
$$\leq 2\|\widetilde{r} - \bar{r}\|_{\widehat{\Sigma}}^2 + c\frac{SA\log\frac{n}{\delta}}{n}\|\widetilde{r} - \bar{r}\|_2^2, \tag{31}$$

in which the inequality is due to the following lemma paraphrased from [37]:

**Lemma 8** (Lemma 7 in [37])**.** For any $\delta \in (0, 1)$, Algorithm 2 satisfies

$$\|\widetilde{r} - \bar{r}\|_\Sigma^2 \leq 2\|\widetilde{r} - \bar{r}\|_{\widehat{\Sigma}}^2 + c^2\frac{SA\log\frac{n}{\delta}}{n}\|\widetilde{r} - \bar{r}\|_2^2 \tag{32}$$

with probability at least $1 - \delta$. $\widehat{\Sigma}$ is the empirical covariance matrix defined in Algorithm 2. $\Sigma$ is the population covariance matrix, i.e., $\Sigma = \mathbb{E}[\widehat{\Sigma}]$. $c > 0$ is an absolute constant.

To proceed with (31), we invoke Lemma 3.1 in [77] for an upper bound on $\|\widetilde{r} - \bar{r}\|_{\widehat{\Sigma}}$, which gives

$$\|\widetilde{r} - \bar{r}\|_\Sigma \leq \sqrt{C^2 R^2 \frac{SA + \log\frac{1}{\delta}}{\gamma^2 n} + c^2\frac{SA\log\frac{n}{\delta}}{n}\|\widetilde{r} - \bar{r}\|_2^2}$$
$$\leq CR\sqrt{\frac{SA + \log\frac{1}{\delta}}{\gamma^2 n}} + c\sqrt{\frac{SA\log\frac{n}{\delta}}{n}}\|\widetilde{r} - \bar{r}\|_2$$
$$\leq CR\sqrt{\frac{SA + \log\frac{1}{\delta}}{\gamma^2 n}} + cR\sqrt{\frac{S^2 A^2 \log\frac{n}{\delta}}{n}},$$

where the last step is because $\|\widetilde{r} - \bar{r}\|_2 \leq 2R\sqrt{SA}$. $C > 0$ is an absolute constant.

## I PROOF OF THEOREM 4

We consider a simple bandit with one state $s$ and two actions $\{a_1, a_2\}$. Since there is only one state in this bandit, we omit the state notation and use the shorthand $(a)$ for $(s, a)$. The true reward of this bandit satisfies $0 \leq r(a_1) < r(a_2) \leq 1$. By Condition 1 of our human rating model, the biased reward also satisfies $0 \leq h(r(a_1)) < h(r(a_2)) \leq 1$.

Without loss of generality, let a human preference feedback $\widetilde{y} = \{0, 1\}$ for this bandit instance be in the following form: $\widetilde{y}_i = 1$ if the human annotator prefers $a_2$, and $\widetilde{y}_i = 0$ if the human annotator prefers $a_1$.

For the preference-based method, the MLE objective is to obtain a reward estimate $f$, or two scalar estimates $f(a_1)$ and $f(a_2)$ that maximizes

$$\sum_{i=1}^n \log\left(\frac{\mathbb{1}\{\widetilde{y}_i = 1\}\exp(f(a_2))}{\exp(f(a_1)) + \exp(f(a_2))} + \frac{\mathbb{1}\{\widetilde{y}_i = 0\}\exp(f(a_1))}{\exp(f(a_1)) + \exp(f(a_2))}\right).$$

We can see that the problem is equivalent to finding a scalar $r(a_2) - r(a_1)$ that maximizes

$$\sum_{i=1}^{n} \log\left(\frac{\mathbb{1}\{\widetilde{y}_i = 1\}}{1 + \exp(-(f(a_2) - f(a_1)))} + \frac{\mathbb{1}\{\widetilde{y}_i = 0\}}{1 + \exp(f(a_2) - f(a_1))}\right).$$

This is also equivalent to the MLE for the parameter $p := \frac{1}{1 + \exp(-(\bar{h}(r(a_2)) - \bar{h}(r(a_1))))}$ of a Bernoulli distribution with samples $\{\widetilde{y}_i\}_{i=1}^{n}$. We know that the solution of this Bernoulli MLE is $\frac{1}{n}\sum_{i=1}^{n}\widetilde{y}_i$.

To identify the optimal arm $a_2$ correctly, the estimated parameter needs to be greater than $\frac{1}{2}$ to show a preference for $a_2$. In the following, we can compute a threshold for the number of samples $n$ such that the failure probability (the algorithm mistakenly identifies $a_1$ as the optimal arm) is at most $\delta$.

$$\mathbb{P}\left[\frac{1}{n}\sum_{i=1}^{n}\widetilde{y}_i \leq \frac{1}{2}\right] \leq \delta$$

This can be done by invoking the Chernoff bound (Lemma 4), which is tight in most cases. Specifically, we can invoke (14) in Lemma 4 with $\epsilon = 1 - \frac{1}{2p}$ and solve for $n$. Thus, the minimum number of samples needed for the preference-based approach to identify the optimal policy $n_{\text{pref}}$ is:

$$n_{\text{pref}} = \frac{2\log\frac{1}{\delta}}{p\epsilon^2} = \frac{2\log\frac{1}{\delta}}{p(1 - \frac{1}{2p})^2}, \tag{33}$$

which implies we can identify the optimal arm correctly with probability at least $1 - \delta$ once the number of samples exceeds this threshold.

For the human rating case, let us denote the human rating data with two sets: $\{\widetilde{r}_i^{(1)}\}_{i=1}^{n_1}$ is the reward ratings for $a_1$, and $\{\widetilde{r}_i^{(2)}\}_{i=1}^{n_2}$ is the reward ratings for $a_2$. Recall for any $a$, $\widetilde{r}(a)$ denotes the empirical average for the reward of $a$ based on the rating data, and $\widehat{r}(a)$ denotes the pessimistic estimate for the reward of $a$ based on $\widetilde{r}(a)$. Notice when the data coverage is uniform, pessimism no longer plays any role in the algorithm.

In this setting, the estimated reward for $a_1$ and $a_2$ needs to satisfy $\widehat{r}(a_1) < \widehat{r}(a_2)$ in order to identify the optimal arm $a_2$ correctly. We want to find the number of samples needed so that the failure probability is controlled below probability $\delta$.

$$\begin{aligned}
&\mathbb{P}\left[\widehat{r}(a_1) \geq \widehat{r}(a_2)\right] \\
&= \mathbb{P}\left[\widetilde{r}(a_1) \geq \widetilde{r}(a_2)\right] \\
&= \mathbb{P}\left[\widetilde{r}(a_1) - \bar{h}(r(a_1)) \geq \widetilde{r}(a_2) - \bar{h}(r(a_1))\right] \\
&= \mathbb{P}\left[\widetilde{r}(a_1) - \bar{h}(r(a_1)) \geq \widetilde{r}(a_2) - \bar{h}(r(a_2)) + \bar{h}(r(a_2)) - \bar{h}(r(a_1))\right] \\
&= \mathbb{P}\left[\frac{1}{n_1}\sum_{i=1}^{n_1}\widetilde{r}_i^{(1)} - \bar{h}(r(a_1)) \geq \frac{1}{n_2}\sum_{j=1}^{n_2}\widetilde{r}_j^{(2)} - \bar{h}(r(a_2)) + \bar{h}(r(a_2)) - \bar{h}(r(a_1))\right] \\
&\stackrel{(i)}{=} \mathbb{P}\left[\frac{1}{n_1}\sum_{i=1}^{n_1}\widetilde{r}_i^{(1)} - \bar{h}(r(a_1)) + \frac{1}{n_2}\sum_{j=1}^{n_2}\widetilde{r}_j^{(2)} - \bar{h}(r(a_2)) \geq \bar{h}(r(a_2)) - \bar{h}(r(a_1))\right]
\end{aligned} \tag{34}$$

To bound the probability in (34) with $\delta$, we can invoke Hoeffding's inequality (Lemma 1) and solve for $n_1$ and $n_2$, which gives

$$\frac{1}{\frac{1}{n_1} + \frac{1}{n_2}} \geq \frac{\sigma^2\log\frac{1}{\delta}}{2\left(\bar{h}(r(a_2)) - \bar{h}(r(a_1))\right)^2}. \tag{35}$$

Recall $n_1 = n_2$. Thus, the minimum number of samples needed for the rating-based approach to identify the optimal policy $n_{\text{rate}}$ is:

$$n_{\text{rate}} = \frac{\sigma^2\log\frac{1}{\delta}}{\left(\bar{h}(r(a_2)) - \bar{h}(r(a_1))\right)^2}.$$

Finally, we have

$$\frac{n_{\text{rate}}}{n_{\text{pref}}} = \frac{\sigma^2 p(1 - \frac{1}{2p})^2}{2\left(\bar{h}(r(a_2)) - \bar{h}(r(a_1))\right)^2},$$

where $p = \frac{1}{1+\exp(-(\bar{h}(r(a_2))-\bar{h}(r(a_1))))}$.

Note $\frac{n_{\text{rate}}}{n_{\text{pref}}}$ is monotonically decreasing for $\bar{h}(r(a_2)) - \bar{h}(r(a_1)) \in [0,1]$ and $\frac{n_{\text{rate}}}{n_{\text{pref}}}$ approaches $0.25\sigma^2$ as $\bar{h}(r(a_2)) - \bar{h}(r(a_1))$ approaches $0$, so we can obtain the advertised result.