# OpenReview forum: "On Provable Benefits of Policy Learning from Human Preferences in Contextual Bandit Problems"
_ICLR.cc/2024/Conference — Submitted to ICLR 2024_

### Official Review · Reviewer_5Lzq · 2023-10-30

**Soundness:** 3 good
**Presentation:** 2 fair
**Contribution:** 2 fair
**Rating:** 3
**Confidence:** 3

**Summary:**

This paper develops a theoretical comparison between these human feedback approaches in offline contextual bandits and shows how human bias and uncertainty in feedback modeling can affect the theoretical guarantees of these approaches. The proposed results seek to provide a theoretical explanation for the empirical successes of preference-based methods from a modeling perspective.

**Strengths:**

1.	The studied problem, i.e., contextual bandits with human feedback, is very well-motivated and finds important applications such as large language models.
2.	The authors propose algorithms based on pessimism with suboptimality guarantees.

**Weaknesses:**

1.	It seems that the proposed algorithms are designed based on standard techniques, such as pessimism and MLE. The authors should elaborate more on their technical novelty. This is my main concern.
2.	It would be more clear to present conditions 1, 2 and 3 as assumptions. The authors should justify more on these assumptions. For example, why is condition 1 reasonable? Why the noise never changes the human preference?
3.	In Theorem 1, the setup $C^*=2$ seems too specific. Can the result be extended to the one that allows general $C^*$ and depends on $C^*$?

**-------After Rebuttal-------**

Thank the authors for their rebuttal. I read the authors' response and other reviewers' comments.

I agree with the comments of Reviewer jnsn, i.e., the logic of this paper is not very reasonable. Specifically, the authors prove that the LCB algorithm with *biased* rating feedback is not sample efficient in Section 4, the pessimistic MLE algorithm with *unbiased* human feedback is sample efficient in Section 5.1, and furthermore, the pessimistic MLE algorithm with *biased* human feedback is also not sample efficient in Section 5.2. Then, without any experiments on the studied settings and designed algorithms, the authors directly come to a conclusion --- the reason that human feedback works well in practical LLMs is probably due to less bias. I do not think the theoretical results in this paper prove (or provide sufficient supports for) this conclusion.

In my opinion, the theoretical analysis in this paper is standard in the bandit and offline RL literature. The authors replied that the main purpose of this paper is to propose a theoretical explanation for the empirical phenomenon. However, with the current theoretical results under the restrictive and biased settings and assumptions, and without any experiments to connect with their settings and algorithms, I do not think the presented theory can effectively explain why human feedback works well in practical LLMs.

Currently I tend to keep my score 3, and will listen to the opinions of other reviewers and AC.

**Questions:**

Please see the weaknesses above.

---

> ### Author Response · Authors · 2023-11-16
> **Reply to Reviewer 5Lzq**
>
> We truly appreciate your comments and questions. Let us answer the questions in your review as follows:
>
> > It seems that the proposed algorithms are designed based on standard techniques, such as pessimism and MLE. The authors should elaborate more on their technical novelty. This is my main concern.
>
> We appreciate you being clear about your main concern. In fact, the goal of this paper is **not** to propose new algorithms but to propose a theory that seeks to explain the superior performance of preference-based methods that people observe in practice. We theorize this is because human rating feedback is less robust to bias than preference feedback, so we propose a new model for it, which we believe is closer to what actually happens in practice than existing models. New theoretical results for these classic algorithms are derived under our new observation model. (As can be seen, this model considers a new type of bias and noise different from the existing results in the literature, and the standard LCB algorithm behaves quite differently under our new model. For example, it would normally converge under partial coverage defined in Assumption 1, but we have shown in Section 4.1 it would not converge under our model.)
>
> > It would be more clear to present conditions 1, 2 and 3 as assumptions. The authors should justify more on these assumptions. For example, why is condition 1 reasonable? Why the noise never changes the human preference?
>
> You’re correct that Conditions 1-3 are assumptions on the human rating model. We can make it clearer in our revision. Your question about Condition 1 is an important one. It is more thoroughly discussed in the paragraph starting with “Apparently” in the middle of Page 4. In short, Condition 1 is necessary because it keeps the estimation problem statistically consistent or well-defined, i.e., there exists an estimator that converges to the optimal policy in probability. In fact, this is an important contribution of our human rating model in Equation (3) over the ones in the existing literature, which do not always admit a solvable problem. In addition, Condition 1 is in alignment with the assumption of the BTL model (human preference) that humans can rank two state-action pairs correctly in expectation, making the comparison between human rating and preference fairer, as explained in Remark 1 on Page 4.
>
> As to why the noise never changes the human preference, we suppose you’re asking why the bias is not in the BTL model for human preference. This is indeed an important question in our theory. It might seem artificial at first to consider bias in the model for human ratings while keeping the BTL model for human preferences unaffected by bias. This is why we include Section 5, in which we consider bias in the BTL model too and find that using human preferences does not have a better sample efficiency than using human ratings in this setting, contrary to what happens in practice. Section 5 shows the superior performance of the human preference approach does not happen because its algorithm is more statistically efficient than the algorithm used with human ratings (standard LCB), but it is likely that this performance difference is caused by a difference in model assumptions between the two approaches (i.e., one type of model/data is easier than the other). This justifies why we theorize the bias affects human ratings but not so much human preferences. We will be happy to strengthen this discussion in our revision.
>
> > In Theorem 1, the setup $C^\star = 2$ seems too specific. Can the result be extended to the one that allows general $C^\star$ and depends on $C^\star$?
>
> This is a good point. $C^\star = 2$ is actually not crucial in our lower bound construction; it can be any constant strictly greater than 1 and would only affect the final suboptimality by a constant factor. In fact, $C^\star = 2$ implies the data have very good coverage (the distribution shift is no more than a factor of 2 over all state-action pairs). The algorithm would only incur a worse constant lower bound under distribution shift with larger $C^\star$, which proves our point even better. Since the goal of this theorem is just to establish the suboptimality is a constant and does not decay as sample size increases, rather than to delineate a specific suboptimality rate, the current presentation is sufficient to demonstrate our point.
>
> Please let us know if this reply is able to address your concerns, and don’t hesitate to let us know if you have any other questions or if you have any suggestions about revision. We look forward to your reply. Thank you!

---

### Official Review · Reviewer_jnsn · 2023-10-31

**Soundness:** 3 good
**Presentation:** 3 good
**Contribution:** 3 good
**Rating:** 3
**Confidence:** 4

**Summary:**

This paper studies the benefits of using preference for learning reward functions in contextual bandits. Through a theoretical analysis for offline contextual bandits, the paper examines how human biases and uncertainties affect these methods' theoretical guarantees. They provided a theoretical explanation for the empirical success of preference-based methods.

**Strengths:**

I found this paper easy to follow. All theoretical conditions and results are clearly stated, and sufficient remarks are followed.

The way that they study the biased reward (through the definition of a transformation $h$) is interesting.

**Weaknesses:**

My biggest concern is that I found the logic of this paper a bit confusing. The authors first showed that LCB failed to achieve the desired statistical guarantee under the biased model (section 4). Then, they showed that pessimism MLE can achieve better statistical results under an unbiased model (section 5.1). This comparison is clearly unfair. Hence, the authors further studied learning preference from the biased model (section 5.2) and showed that the results are actually worse. They then remarked that

> This shows if one assumes a similar amount of human bias and uncertainty in both types of human feedback, the preference-based approach is no more sample-efficient. This actually contradicts with the empirical observations in the existing literature, which suggests preference-based methods have superior performance. Hence, our theory shows the bias-free modeling plays a great role in the lower sample complexity of preference-based methods, and our theoretical results can conversely confirm the standard BTL modeling of human preference feedback—it is reasonable to believe human preference data is indeed subject to less bias and uncertainty in practice.

My understanding is that, the authors are not trying to use *theory* to verify *empirical success* (which I was expecting), but rather, they use *empirical success* to prove the *theory*. Hence, it appears that this paper has undertaken a completely contrary endeavor. The authors seem to haven't truly shown any benefits of using preference from pure theory; on the contrary, the conclusions they have drawn are rather contradictory (Theorem 4). Their sole argument positing the superiority of preference relies on the fact that, in practice, it yields better experimental results, thereby suggesting that the preference is unlikely to be significantly biased. If it is really what the authors intended to convey, I don't think this result is a "provable" benefit but rather heuristic. This leaves me quite confused, and I hope the authors can clarify this point.



Some other issue: the lower bound results (theorem 1 & 2) only considered the LCB algorithm. It will be more convincing to establish a universal and information-theoretic lower bound, i.e., a lower bound that holds for any algorithm.

**Questions:**

Is the studied algorithm, pessimistic MLE, computationally efficient? If it is not, I don't think it is fair to compare it with the more efficient LCB algorithm. Actually this question circles back to the previous one: can the lower bound be applicable to any algorithm and not solely limited to LCB?

---

> ### Author Response · Authors · 2023-11-16
> **Reply to Reviewer jnsn (Part 1)**
>
> We truly appreciate your comments and your thorough review. Let us answer the questions in your review as follows:
>
> > My biggest concern is that I found the logic of this paper a bit confusing…
>
> > My understanding is that, the authors are not trying to use theory to verify empirical success (which I was expecting), but rather, they use empirical success to prove the theory…
>
> Yes, this is a good question. Overall, what this paper tries to do is propose a theory that can explain a phenomenon in practice. The logic of this paper is intended to be as follows:
>
> Learning with human preference data is observed to yield much better performance in practice, compared to prior efforts of learning with human ratings. It still remains a question what actually causes this superior performance. Thus, we seek to explain this phenomenon by proposing a theory: human ratings are actually subject to more human bias in practice, similar to the model we propose in Section 3, while the generation of human preferences is more robust to such bias (the BTL model that people currently use is already faithful to the real world). It has been a common belief that it is more accurate for human annotators to compare than to rate directly [1,2], but granted, it might seem unfair at first to consider bias in the model for human ratings while keeping the BTL model for human preferences unaffected by bias. We justify this in Section 5, in which we consider bias in the BTL model too and find that using human preferences does not have a better sample efficiency than using human ratings in this setting, contrary to what happens in practice. This shows the superior performance of the human preference approach does not happen because its algorithm (pessimistic MLE) is more statistically efficient than the algorithm used for human ratings (standard LCB); this performance difference is more likely caused by a difference in model assumptions between the two approaches (i.e., one type of model/data is easier than the other).
>
> Overall, the goal of this paper is to propose a reasonable theory for a real-world phenomenon. It would be possible to “use theory to verify empirical success” only if the mathematical formulation and underlying assumptions of the problem of interest are clear. However, it is not the case in this problem, since it is unclear what causes the human preference approach to be better in practice. (Clearly the basic model with subgaussian random reward from the theoretical literature falls short of explaining this.) For this reason, we think it is more appropriate to propose a (heuristic) theory and make sure it is in line with the real-world phenomenon and is also justifiable.
>
> [1] Touvron, Hugo, et al. "Llama 2: Open foundation and fine-tuned chat models." arXiv preprint arXiv:2307.09288 (2023).
>
> [2] Kevin R. Tarlow, Daniel F. Brossart, Alexandra M. McCammon, Alexander J. Giovanetti, M. Camille Belle, and Joshua Philip. Reliable visual analysis of single-case data: A comparison of rating, ranking, and pairwise methods. Cogent Psychology, 8(1):1911076, 2021. doi: 10.1080/23311908.2021.1911076.
>
> > Some other issue: the lower bound results (theorem 1 & 2) only considered the LCB algorithm. It will be more convincing to establish a universal and information-theoretic lower bound, i.e., a lower bound that holds for any algorithm.
>
> This is a good suggestion. Theorem 2 can be made into a universal lower bound, but Theorem 1 is specific to algorithms that use pessimism. In fact, we tailor these lower bounds to specific algorithms because together with Section 5, it rules out any algorithmic factor that causes the human preference approach to perform better. It serves our message better to keep the algorithm in the picture. If we proved an info-theoretic lower bound, the message would become “the human rating data following the model in Equation (3) are fundamentally harder than human preference data following the BTL”. Although it still is something we want to show, it no longer has the additional benefit of ruling out the algorithmic factor.

---

> > ### Author Response · Authors · 2023-11-16
> > **Reply to Reviewer jnsn (Part 2)**
> >
> > > Is the studied algorithm, pessimistic MLE, computationally efficient? If it is not, I don't think it is fair to compare it with the more efficient LCB algorithm. Actually this question circles back to the previous one: can the lower bound be applicable to any algorithm and not solely limited to LCB?
> >
> > Pessimistic MLE can be implemented in practice, as the paper that introduces it has shown [1]. In particular, we focus on the tabular setting, so it is computationally efficient. It is currently regarded as the standard algorithm in the preference feedback offline RL/bandit literature as LCB is in the basic offline RL/bandit literature.
> >
> > [1] Banghua Zhu, Jiantao Jiao, and Michael I Jordan. Principled reinforcement learning with hu- man feedback from pairwise or k-wise comparisons. arXiv preprint arXiv:2301.11270, 2023.
> >
> > Please let us know if this reply is able to address your concerns, and don’t hesitate to let us know if you have any other questions or if you have any suggestions about revision. We look forward to your reply. Thank you!

---

> ### Comment · Reviewer_jnsn · 2023-11-22
>
> Thanks for the detailed response! I really appreciate that the authors went through the logic of the paper again in detail. However, I still can't find the logic of the paper to make sense. If my understanding is correct, the theoretical conclusion of this paper is still not drawn from theoretical derivation but from empirical observation, as the author said in response
>
> > ... this performance difference is more likely caused by a difference in model assumptions between the two approaches ...
>
> To be more specific, the authors equivalently did the following in sequence: (1) proposed some theoretical assumptions (the biased ones), (2) did some derivation, (3) observed that the theoretical results don't match the empirical performance, (4) claimed that this mismatch is due to a wrong assumption that they started with, and finally (5) turned to the other assumption (the bias-free one) and claimed that this explains the benefit of learning from preference.
>
> I still don't think this logic is convincing. In my opinion, the more correct and common way to show the provable benefit of something is the following: (1) propose some assumptions and **some justification for the assumption**, (2) do some derivation and get some results, and (3) observe that the theoretical results match the empirical performance. By comparing this logic with that of the paper, I found that this paper doesn't have an explicit part that shows **some justification for the assumption**. Instead, the way that they justify the assumptions is by empirical observation. In particular, they justify the "bias-free" assumption by the empirical success, which is exactly what they need to theoretically justify. So I feel there is a **circular reasoning**.
>
> Moreover, the fact that the paper failed to show the superiority of learning from preference under equal assumptions (ie, the biased one) is unnecessarily due to that learning from preference is bias-free. Actually, this concluded assumption seems super strong in the sense that humans can't be bias-free anyway (but the authors removed the "bias" in the formulation in section 5.1, although I admit that the BTL has some randomness itself). I would suggest the authors to try some more refined assumptions or analyses instead of "bias" vs "bias-free", which also seems to trivialize the problem in the sense that "bias-free" should intuitively be better.
>
> Again I greatly appreciate the detailed response by the authors. However, I still fail to see the rationale of this work. Please correct me if I missed any key parts. I still can't raise my assessment of this work for now since my biggest concern still exists.

---

> > ### Author Response · Authors · 2023-11-23
> > **Reply to Reviewer jnsn**
> >
> > Thank you for the reply! We really appreciate you being clear about your concerns. Let us first provide some further clarification for your question about “the justification for our assumption”. We understand you still have doubts about us justifying our biased rating model with whether our final results match the empirical observations. In fact, there is another important reason why we propose this rating model, as we have noted in our paper and last response, which is the fact that it has been long hypothesized that human annotators can give more accurate feedback when asked to compare than when asked to rate directly. Evidence of such belief can be found in the “Beyond Human Supervision” paragraph of Section 5.1 in the Llama 2 paper [1], prior efforts of studying human rating (non-bandit) [2,3], and psychology papers like [4]. Works like [2,3] have already attempted to formulate the bias in human rating, but a huge limitation is that none of these models guarantees the statistical consistency of the resulting policy learning problem (as we describe in Section 3). Overall, there is plenty of support for our consideration of bias in human rating from the existing literature.
> >
> > In addition, we would also like to take the opportunity to explain a bit more about the importance of our current Section 5.2, which essentially shows equal bias (between the two approaches) implies no better performance for human preference, as you have noted. The contrapositive of this is “better performance of human preference implies unequal bias”. This can be presented as another reasonable justification for our model at the very beginning and should have no circular dependency in logic.
> >
> > Thus, our assumption that human rating feedback contains more bias is grounded. To match the logic flow that you suggested in the latest reply, we can revise our paper by first presenting the two justifications above before proposing our biased model for human rating (from Section 3). Then we will provide our theoretical results, observe they match the empirical performance, and conclude our theory is a reasonable explanation to the empirical observations.
> >
> > On the other hand, we also hope to provide our thoughts on your question about “a more refined assumption for bias vs bias-free”. This is a good question. Since the $h$ transformation is allowed to be any general function satisfying Condition 1-3, our current model can just be viewed as a general framework that can express any more refined bias and one can just plug in the more refined $h$ into our current theoretical results directly. Nonetheless, we admit that the current assumption could be improved. We hope the reviewer could kindly consider that this is only the first work that studies the problem of human comparison vs human rating in the RLHF context, given RLHF is a nascent field with incomplete theoretical understanding. We expect there to be future works to further investigate this problem and improve the modeling of bias.
> >
> > We hope this could dispel some of your concerns. Finally, as the discussion period is coming to an end, we would like to thank you again for taking the time to review our submission!
> >
> > [1]  Touvron, Hugo, et al. "Llama 2: Open foundation and fine-tuned chat models." arXiv preprint arXiv:2307.09288 (2023).
> >
> > [2] Marıa Pérez-Ortiz, Aliaksei Mikhailiuk, Emin Zerman, Vedad Hulusic, Giuseppe Valenzise, and Rafał K. Mantiuk. From pairwise comparisons and rating to a unified quality scale. IEEE Transactions on Image Processing, 29:1139–1151, 2020.
> >
> > [3] Zhi Li, Christos G Bampis, Luka ́sˇ Krasula, Lucjan Janowski, and Ioannis Katsavounidis. A simple model for subject behavior in subjective experiments. arXiv preprint arXiv:2004.02067, 2020.
> >
> > [4] Kevin R. Tarlow, Daniel F. Brossart, Alexandra M. McCammon, Alexander J. Giovanetti, M. Camille Belle, and Joshua Philip. Reliable visual analysis of single-case data: A comparison of rating, ranking, and pairwise methods. Cogent Psychology, 8(1):1911076, 2021. doi: 10.1080/23311908.2021.1911076.

---

### Official Review · Reviewer_TtXD · 2023-11-01

**Soundness:** 4 excellent
**Presentation:** 4 excellent
**Contribution:** 3 good
**Rating:** 8
**Confidence:** 3

**Summary:**

This paper analyzes the effect of bias in human feedback for contextual bandits in two settings: rating feedback (i.e. direct access to the reward function), and preference (comparison) feedback.

The effect of human biased is first quantified by proving sub-optimality bounds of two (previously existing) algorithms for contextual bandits with human feedback. The novelty in the presented bounds lie in the fact that feedback (rating or preference) is received through a bias transform.  It is later shown than for a certain class of bias transformations, solving a bandit problem with biased rating feedback, always requires more samples than solving the same problem with biased preference feedback. This is an interesting phenomena as works against the prior conception that solving a bandit/RL problem with preference feedback is more complex than with rating feedback.
At the same time, this statement is not entirely surprising, since the BTL preference feedback model is robust to the considered class of bias transformations.

**Strengths:**

My summary should reflect some strengths of the paper, I spell out a few more below.

* The paper is well written and straightforward.

* It tackles an important problem and gives a clear answer. In particular, Theorem 4, which is algorithm independent, has a really nice formulation.

* I am not aware of prior work on Bandits/RL with human rating to assess how novel Theorem 1 is compared to previous sub-optimality of the LCB algorithm (e.g. in terms of techniques used to derive it). However, Theorem 1 and 3 clearly characterize the effect of a transformed/biased rating feedback.

**Weaknesses:**

* Given the assumptions on the bias transform $h$, I am not surprised that the preference-based feedback is rather invariant to such biases. So I am not sure if the final results are uncovering an informative phenomena.

* A number of prior works on contextual bandits with preference feedback is not mentioned. While the overall approach is sufficiently different (they optimize a least squared loss), I think they should be mentioned for completeness.
  - Mehta, Viraj, et al. "Kernelized Offline Contextual Dueling Bandits." arXiv preprint arXiv:2307.11288 (2023).
  - Dudík, Miroslav, et al. "Contextual dueling bandits." Conference on Learning Theory. PMLR, 2015.
  - Saha, Aadirupa, and Akshay Krishnamurthy. "Efficient and optimal algorithms for contextual dueling bandits under realizability." International Conference on Algorithmic Learning Theory. PMLR, 2022.
- Bengs, Viktor, Aadirupa Saha, and Eyke Hüllermeier. "Stochastic Contextual Dueling Bandits under Linear Stochastic Transitivity Models." International Conference on Machine Learning. PMLR, 2022.
- Perhaps also: Bengs, Viktor, et al. "Preference-based online learning with dueling bandits: A survey." The Journal of Machine Learning Research 22.1 (2021): 278-385.

**Questions:**

- How would you go beyond tabular setting? Would you say the pessimistic MLE algorithm can be easily extended to say, a kernelized or linear rewards over a compact domain?

---

> ### Author Response · Authors · 2023-11-16
> **Reply to Reviewer TtXD**
>
> We truly appreciate your comments and support. Let us answer the questions in your review as follows:
>
> > Given the assumptions on the bias transform $h$, I am not surprised that the preference-based feedback is rather invariant to such biases. So I am not sure if the final results are uncovering an informative phenomena.
>
> Your question is completely valid. The theoretical results are not meant to be surprising. There are two reasons for this. First, it is already a common belief that it is more accurate for human annotators to give preference feedback than to give rating feedback [1,2], but there hasn’t been any work that formalizes this theoretically. Furthermore, it is also notable that the bias in our rating model is monotone and does not change the optimal policy of the problem, so it is actually not immediately obvious why such bias makes learning from human ratings harder. Second, while it is not surprising that the robustness of human preferences to bias is a factor of why preference-based methods perform better in practice, we make an important contribution by pinpointing this factor as the likely primary cause among other factors. Specifically, we rule out the algorithmic factor by showing the pessimistic MLE used in preference-based approach is no more statistically efficient than the standard LCB used in rating-based approach.
>
> [1] Touvron, Hugo, et al. "Llama 2: Open foundation and fine-tuned chat models." arXiv preprint arXiv:2307.09288 (2023).
>
> [2] Kevin R. Tarlow, Daniel F. Brossart, Alexandra M. McCammon, Alexander J. Giovanetti, M. Camille Belle, and Joshua Philip. Reliable visual analysis of single-case data: A comparison of rating, ranking, and pairwise methods. Cogent Psychology, 8(1):1911076, 2021. doi: 10.1080/23311908.2021.1911076.
>
> > A number of prior works on contextual bandits with preference feedback is not mentioned. While the overall approach is sufficiently different (they optimize a least squared loss), I think they should be mentioned for completeness.
>
> Thank you for providing these references! We will include them in our revision.
>
> > How would you go beyond tabular setting? Would you say the pessimistic MLE algorithm can be easily extended to say, a kernelized or linear rewards over a compact domain?
>
> There is no problem extending pessimistic MLE to a more general function approximation, as it has been studied in the linear function approximation setting by [1] and the general function approximation setting by [2]. However, how to extend our newly proposed rating model to the linear setting still needs investigation. Recall our model considers a general bias that is likely nonlinear in the real world. While a tabular algorithm is able to learn such biased reward, linear function approximation would suffer a huge approximation error; RKHS and more general function approximation like neural networks should be flexible enough to approximate such biased reward.
>
> [1] Banghua Zhu, Jiantao Jiao, and Michael I Jordan. Principled reinforcement learning with hu- man feedback from pairwise or k-wise comparisons. arXiv preprint arXiv:2301.11270, 2023.
>
> [2] Wenhao Zhan, Masatoshi Uehara, Nathan Kallus, Jason D Lee, and Wen Sun. Provable offline reinforcement learning with human feedback. arXiv preprint arXiv:2305.14816, 2023.
>
> Please don’t hesitate to let us know if you have any further questions. Thank you!

---

### Meta-Review · Area_Chair_17bt · 2023-12-07

**Metareview:**

This paper proposes a theory for why reward function approximation from _human preference feedback_ (i.e., human says "I like $A$ more than $B$") often outperforms reward function approximation from _human rating feedback_ (i.e., human says "I value $A$ as $X$"). The proposed theory suggests that _bias_ is the primary culprit; more precisely, that human ratings are subject to bias, while human preferences are not. To demonstrate this, the authors prove the following:

1. Under a very general bias model (proposed in the paper), the LCB algorithm (for learning a policy from collected human ratings) has a constant suboptimality gap; meaning, it cannot asymptotically recover the optimal policy. (This result is proven two ways, under assumptions that the logging policy has either common support with the target policy, or full support.)
2. Under the BTL preference model _with no bias_, the pessimistic MLE algorithm (for learning a policy from collected human preferences) has asymptotically decaying suboptimality; meaning, it will eventually recover the optimal policy.
3. Under the BTL preference model _with bias_, for a single-state (effectively, non-contextual) bandit problem with 2 actions, pessimistic MLE has sample complexity that is no better than LCB, up to a constant multiplicative factor.

From these three results they deduce that the distinguishing characteristic for why preference-based learning outperforms rating-based learning must be the bias.

All reviewers agreed that the paper is very well written; and to the best of their knowledge, the theorems are sound. That being said, the main point of contention was whether the paper's overall reasoning is sound; whether the main hypothesis -- that preference-based learning is better due to lack of (or less) bias -- actually follows from the theoretical findings.

The logic is somewhat like an ablation study that goes something like the following. Under bias, prove that LCB is bad. Without bias, prove that pessimistic MLE is good. Then, with bias, prove that pessimistic MLE is bad (or no better than LCB). Therefore, varying the bias determines whether pessimistic MLE is good or bad; and since pessimistic MLE performs well in practice, it must be due to bias.

At first blush, this reasoning made sense to me; but the more I thought about it, and with help/discussion from reviewers `jnsn` and `5Lzq`, I have come around to see the flaws in this reasoning. The fact that pessimistic MLE has no advantage (over LCB) in the presence of bias does not necessarily mean that bias is the primary culprit. As `jnsn` points out, it could be due to inappropriateness of the BTL model or the chosen algorithm. Viewed as such, it seems like a stretch to claim that preference-based learning's empirical dominance must be due to a lack of bias.

I think this is an incredibly interesting and relevant paper for the ICLR community -- especially these days -- and I applaud the authors for tackling it. This is solid work; it seems like a good start... but I just don't think that the paper, in its current form, is ready. The logical reasoning needs to be refined or presented in a way that is more convincing.

**Justification For Why Not Higher Score:**

It is possible that I, and the other reviewers, have misunderstood the paper. But if that's the case, that 2 reviewers and an AC misunderstood the work, then it indicates that the paper did a poor job of explaining its reasoning. If the reasoning is in fact sound, then the authors should revise the paper so that its message is more comprehendable.

**Justification For Why Not Lower Score:**

N/A

---

### Decision · Program_Chairs · 2024-01-16

Reject